# ENTROPY-SELECT: TRAINING-FREE LOCAL ENTROPY TOKEN COMPRESSION FOR VIDEO LLMS

## ABSTRACT

Video Language Models (VLMs) emit visual tokens that grow linearly with video length while attention scales quadratically, making inference sensitive to token count and latency variance. In practice, an effective token compressor should be training-free to plug into strong off-the-shelf VLMs, architecture-agnostic to avoid model-specific hooks, and deliver selection runtime that is predictable regardless of the target retention—properties that many prior methods lack. We introduce ENTROPY-SELECT, a training-free, architecture-agnostic framework that ranks tokens by local neighborhood entropy, an information-theoretic measure of unpredictability relative to nearby tokens. We compute temperature-scaled similarity distributions within fixed spatial windows to obtain normalized entropy, fuse it with gradient-based structural saliency, enforce coverage via grid quotas, and allocate per-frame budgets by saliency. Because windows/grids are fixed and selection reduces to a single global sort, ENTROPY-SELECT runs in $\mathcal{O}(N \log N)$ with latency effectively decoupled from the retention ratio. Across MVBench, ActivityNet-QA, VideoMME, and EgoSchema, ENTROPY-SELECT matches or exceeds the uncompressed baseline at moderate retention and degrades gracefully under aggressive compression, yielding predictable latency and plug-and-play deployment without any model modification or internal-state access. Notably, we observe "enhancement under compression" on several benchmarks—for example, 35% and 50% retention surpass the full-token baseline on benchmarks such as MVBench and EgoSchema—suggesting that removing low-entropy redundancy can improve downstream accuracy.These results establish local entropy as a principled criterion for video token selection and a practical tool for scalable VLM inference. Code will be released upon acceptance.

## 1 INTRODUCTION

Video understanding with large multimodal models is increasingly central to long-horizon perception and reasoning, yet practical deployment remains constrained by a compute mismatch: visual tokens grow linearly with video duration while attention scales quadratically. A single minute of standard-resolution video can produce hundreds of thousands of tokens, quickly exhausting memory and inflating latency, especially when temporal coherence must be preserved alongside spatial detail (Li et al., 2024b; Xu et al., 2024; Zhang et al., 2024a). In production settings, the selector that decides which tokens to keep must also satisfy three deployment desiderata: it should be *training-free* to plug into strong off-the-shelf VLMs, architecture-agnostic to avoid model-specific hooks, and exhibit selection runtime that is predictable—i.e., decoupled from the user-chosen retention ratio (Fig. 1a).

Existing approaches tackle token proliferation along three lines. (i) *Architecture-level compression embeds pooling or specialized modules* (Li et al., 2024f; Xu et al., 2024; Yao et al., 2024; Zhang et al., 2024a); these can optimize the rate–accuracy trade-off but require retraining or projector changes, limiting portability. (ii) *Training-free but internals-dependent selection infers importance from attention/KV/CLS or intermediate features* (Chen et al., 2024b; Yang et al., 2024b; Tao et al., 2025; Tan et al., 2025; Yang et al., 2025; Jeddi et al., 2025; Ouyang et al., 2025; Chen et al., 2025; Zhang et al., 2025); such methods are difficult to deploy on black-box or commercial VLMs and their iterative pruning/graph/recovery pipelines couple latency to the target retention. (iii) *Connectivity/region-driven strategies improve boundary preservation* yet still incur graph costs that

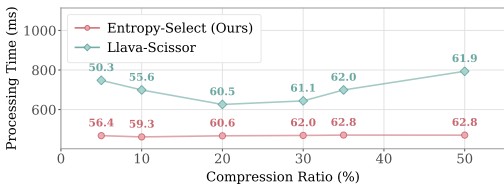 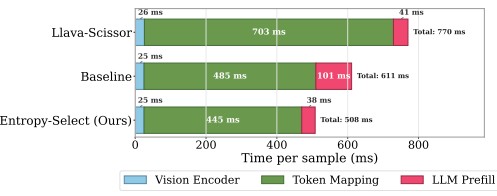

(a) Token compression runtime vs ratio.    (b) Stage-wise end-to-end latency at 35% retention.

Figure 1: Token compression scaling and the 35% retention stage-wise latency breakdown.

scale with sequence length and retention (Sun et al., 2025). Across these lines, two conceptual gaps persist: the motivation for "importance" is often a saliency proxy (attention, similarity, connectivity) rather than an information measure; and the lack of a fixed-complexity stencil yields unpredictable selection latency.

We address these gaps with ENTROPY-SELECT, a training-free, architecture-agnostic token selector grounded in information theory. Our key premise is that a token is valuable if it is hard to predict from its local neighborhood. Concretely, we compute temperature-scaled similarity distributions within fixed spatial windows and use their normalized Shannon entropy as an information score; we fuse this with gradient-based structural saliency, enforce spatial coverage via grid quotas, and allocate per-frame budgets by saliency. Because windows and grids are fixed and selection reduces to a single global sort, the complexity is $\mathcal{O}(N \log N)$ *and effectively decoupled from the retention ratio*. Operating purely on exported features without attention/KV access, ENTROPY-SELECT delivers predictable latency and plug-and-play deployment. As shown in Fig. 1b, our method reduces overall inference time by decreasing token mapping and LLM prefill costs while maintaining modest selector overhead. Experiments on MVBench, ActivityNet-QA, VideoMME, and EgoSchema show that ENTROPY-SELECT matches or surpasses uncompressed accuracy at moderate retention and degrades gracefully under aggressive compression, establishing local entropy as a principled criterion for scalable video token selection.

## 2 RELATED WORK

**Token Compression for VLMs: What Is Missing?** Token compression has motivated three lines of work. Architecture-level compression. Methods embed pooling or specialized modules into the pipeline (Li et al., 2024f; Xu et al., 2024; Yao et al., 2024; Zhang et al., 2024a). They can optimize the rate–accuracy trade-off but require retraining or projector changes, reducing portability to strong off-the-shelf VLMs. Training-free but *internals-dependent* selection. FastV uses prefilling attention (Chen et al., 2024b); VisionZip relies on CLS attention and merging (Yang et al., 2024b); DyCoke leverages temporal grouping and dynamic caches (Tao et al., 2025); recent variants estimate importance via information flow, semantic consistency, or similarity graphs (Tan et al., 2025; Yang et al., 2025; Jeddi et al., 2025; Ouyang et al., 2025; Chen et al., 2025; Zhang et al., 2025). While training-free, these methods typically *assume* access to attention, KV, or intermediate features and their selection costs often scale with the retention ratio through iterative pruning, graph construction, or recovery steps. Connectivity/region-driven selection. LLaVA-Scissor identifies Semantic Connected Components to preserve region integrity (Sun et al., 2025). It improves boundary preservation but still introduces graph costs whose runtime varies with sequence length and target retention. Orthogonally, speculative decoding (Leviathan et al., 2023; Cai et al., 2024; Li et al., 2024) and attention-head sparsity methods (Wang et al., 2025; Kang et al., 2025; Huang et al., 2025) accelerate the generation phase or reduce per-layer attention cost; these are complementary to prefill-stage token compression and could be combined with ENTROPY-SELECT for multiplicative efficiency gains. As shown in Table 1, from a deployment viewpoint, three issues persist: (1) unclear motivation—importance is usually a *proxy* (attention/CLS/graph) rather than an information measure; (2) portability—internal-state access is unavailable in black-box or commercial VLMs; (3) predictability—selection latency couples with the user-defined retention ratio, complicating service-level guarantees.

**Information Theory for Neural Compression.** Information theory provides principled objectives for compression via entropy, mutual information, and rate–distortion (Tishby & Zaslavsky, 2015;

Saxe et al., 2019; Ballé et al., 2016; 2018; Cheng et al., 2020; Achille & Soatto, 2018). Despite progress in codecs and pruning, VLM token selection has largely remained heuristic. Our work instantiates a direct, local-entropy criterion: tokens surrounded by heterogeneous neighborhoods exhibit higher unpredictability and thus higher potential semantic gain. This closes the motivation gap by grounding selection in information content rather than model-specific saliency, and it avoids internal access—operating purely on exported features with fixed windows/grids for predictable complexity.

Table 1: Deployment-critical properties uniquely satisfied by ENTROPY-SELECT.

| Method | Training free | Architecture agnostic | Complexity decoupled |
|---|---|---|---|
| PLLaVA (Xu et al., 2024) | | | |
| VisionZip (Yang et al., 2024b) | ✓ | | |
| DyCoke (Tao et al., 2025) | ✓ | | |
| LLaVA-Scissor (Sun et al., 2025) | ✓ | ✓ | |
| VFlowOpt (Yang et al., 2025) | ✓ | | |
| ENTROPY-SELECT (Ours) | ✓ | ✓ | ✓ |

Table 1 summarizes three deployment properties: Training-free, Architecture-agnostic, and Complexity-decoupled (selection runtime independent of retention). Prior work typically satisfies at most one or two. ENTROPY-SELECT is, to our knowledge, the first to meet all three simultaneously: no retraining, no internal signals, and fixed-window/grid computation plus a single global sort, $\mathcal{O}(N \log N)$, yielding predictable latency regardless of retention.

## 3 METHOD

### 3.1 PROBLEM FORMULATION

Given a video sequence $\mathbf{V} = \{v_1, ..., v_n\}$ with $n$ frames, a vision encoder produces token representations $\mathbf{T} \in \mathbb{R}^{N \times D}$ where $N = n \times m$ represents total tokens ($m$ tokens per frame) with $D$-dimensional features. Our goal is to select a subset $\mathbf{T}' \in \mathbb{R}^{K \times D}$ where $K \ll N$ that maximally preserves information necessary for downstream understanding while minimizing computational cost. We formulate this as an information-theoretic optimization problem: identify tokens whose local neighborhoods exhibit high entropy, indicating diverse and unpredictable information content. Our approach consists of three integrated components: (1) local entropy computation to quantify information diversity within spatiotemporal neighborhoods, (2) gradient-entropy fusion that combines information diversity with structural edges for coverage-aware selection, and (3) saliency-weighted frame allocation that intelligently distributes the compression budget across temporal frames based on their information content.

### 3.2 LOCAL SPATIOTEMPORAL WINDOW ENTROPY

The foundation of our approach is measuring information diversity within spatiotemporal neighborhoods. Tokens are arranged per frame on a 2D grid of height $H$ and width $W$, and we index a token by $(t, r_i, c_i)$ with feature $\mathbf{t}_{t,r_i,c_i} \in \mathbb{R}^D$. For each reference token, we define a 3D neighborhood $\mathcal{N}_{t,r_i,c_i}$ that unions (i) a $k \times k$ spatial window in the current frame $t$ and (ii) $k_{past} \times k_{past}$ spatial windows over a past-only temporal window of depth $K_t$ (time offsets $\Delta t \in \{-(K_t-1), \dots, -1, 0\}$), all centered at $(r_i, c_i)$ and truncated by image boundaries. Ignoring boundary truncation and excluding the center token in the current-frame window, the neighborhood size simplifies to

$$|\mathcal{N}_{t,r_i,c_i}| = (k^2 - 1) + (K_t - 1) k_{past}^2. \tag{1}$$

Pairwise similarities within each local window are computed as $s_{ij} = \hat{\mathbf{t}}_i^T \hat{\mathbf{t}}_j$ for $j \in \mathcal{N}_i$. These similarities are transformed into probabilities using temperature-scaled softmax:

$$p_{ij} = \frac{\exp(s_{ij}/\tau)}{\sum_{\ell \in \mathcal{N}_i \setminus \{i\}} \exp(s_{i\ell}/\tau)}, \tag{2}$$

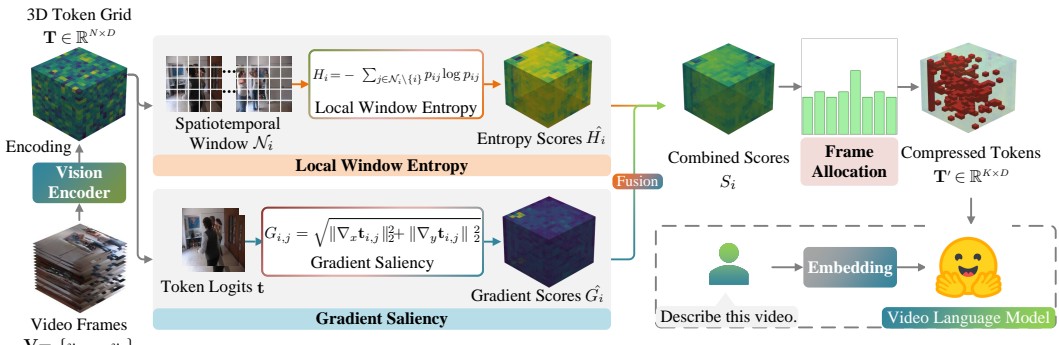

Figure 2: Overview of ENTROPY-SELECT. Video frames are encoded into a 3D token grid, where local entropy and gradient saliency are computed and combined to guide token selection with spatial coverage and frame-level allocation, producing a compressed set that preserves key information.

where temperature $\tau$ controls the entropy of the distribution–lower values create peaked distributions emphasizing strong similarities, while higher values produce more uniform distributions.

The local entropy quantifies neighborhood diversity as $H_i = -\sum_{j \in \mathcal{N}_i \backslash \{i\}} p_{ij} \log p_{ij}$. To enable comparison across different positions, we normalize by the maximum possible entropy: $\hat{H}_i = H_i / \log(|\mathcal{N}_i| - 1) \in [0, 1]$. This normalized entropy directly measures local information content: values approaching 1 indicate tokens in heterogeneous neighborhoods (boundaries, transitions), while values near 0 suggest homogeneous regions (uniform textures).

### 3.3 GRADIENT-ENTROPY FUSION AND COVERAGE-AWARE SELECTION

While entropy captures information diversity, explicit structural features enhance selection quality. For tokens arranged in the 2D grid, we compute spatial gradients to identify edges:

$$G_{i,j} = \sqrt{\|\nabla_x \mathbf{t}_{i,j}\|_2^2 + \|\nabla_y \mathbf{t}_{i,j}\|_2^2}, \tag{3}$$

where $\nabla_x$ and $\nabla_y$ represent horizontal and vertical differences computed between adjacent tokens.

The combined importance score balances information diversity and structural significance: $S_i = \alpha \cdot \hat{H}_i + (1 - \alpha) \cdot \hat{G}_i$, with $\alpha = 0.5$ empirically balancing both contributions and $\hat{G}_i$ being the min-max normalized gradient magnitude.

To prevent over-concentration in high-scoring regions, we enforce spatial coverage through grid partitioning. Each frame is divided into $g \times g$ regions (typically $g = 4$), with minimum token retention per region: $k_{\text{region}} = \max(1, \lfloor B_f \cdot \rho/g^2 \rfloor)$, where $B_f$ is the frame's token budget and $\rho$ is the spatial preservation ratio. Within each grid cell, we select the top-$k_{\text{region}}$ tokens by score, guaranteeing spatial distribution. The remaining budget is allocated globally to the highest-scoring unselected tokens.

### 3.4 SALIENCY-WEIGHTED FRAME ALLOCATION

For multi-frame videos, we allocate compression budgets across frames based on their information content. We compute per-frame token scores and use a softmax-based allocation strategy.

For each frame $f$, we first compute the combined scores for all tokens as described above. Then, we calculate a robust mean score using the top-$q$ fraction of tokens (typically $q = 0.3$): $\bar{S}_f = \frac{1}{|T_q|} \sum_{i \in T_q} S_i$, where $T_q$ represents the set of top-$q$ highest-scoring tokens in frame $f$.

Frame budgets are allocated using temperature-scaled softmax:

$$w_f = \frac{\exp(\bar{S}_f / \tau_{\text{alloc}})}{\sum_{f'=1}^{n} \exp(\bar{S}_{f'} / \tau_{\text{alloc}})}. \tag{4}$$

Table 2: Performance on long video understanding benchmarks and comprehensive tasks. Best results in **bold**, second-best underlined. Avg.(%) is the mean of per-dataset ratios relative to the 100% row.

| Method | RR | EgoSchema | MLVU | VideoMME | MVBench | Avg.(%) |
|---|---|---|---|---|---|---|
| LLaVA-OneVision (Li et al., 2024b) | 100% | 58.08 | 62.48 | 57.96 | 62.43 | 100.0% |
| *Token Retention Ratio = 50%* | | | | | | |
| DyCoke (Tao et al., 2025) | 50% | 57.74 | 61.09 | 57.35 | 61.48 | 98.7% |
| PLLaVA (Xu et al., 2024) | 50% | 57.72 | 61.15 | 56.93 | 60.10 | 98.0% |
| FastV (Chen et al., 2024b) | 50% | 58.00 | 61.27 | 57.47 | 61.28 | 98.8% |
| VisionZip (Yang et al., 2024b) | 50% | 53.57 | 57.03 | 54.19 | 58.28 | 92.6% |
| LLaVA-Scissor (Sun et al., 2025) | 50% | 57.58 | 61.32 | 57.37 | 61.85 | 98.8% |
| ENTROPY-SELECT (Ours) | 50% | **58.12** | **61.50** | **58.26** | **62.77** | **99.9%** |
| *Token Retention Ratio = 35%* | | | | | | |
| DyCoke (Tao et al., 2025) | 35% | 57.74 | 59.95 | 56.22 | 61.78 | 97.8% |
| FastV (Chen et al., 2024b) | 35% | 57.75 | 59.54 | 56.00 | 61.28 | 97.3% |
| PLLaVA (Xu et al., 2024) | 35% | 56.07 | 59.42 | 54.26 | 59.48 | 95.1% |
| VisionZip (Yang et al., 2024b) | 35% | 52.00 | 56.29 | 53.70 | 56.98 | 90.8% |
| LLaVA-Scissor (Sun et al., 2025) | 35% | 57.94 | 60.95 | 57.52 | 61.98 | 99.0% |
| ENTROPY-SELECT (Ours) | 35% | **58.34** | **61.48** | **57.74** | **62.83** | **99.8%** |
| *Token Retention Ratio = 10%* | | | | | | |
| FastV (Chen et al., 2024b) | 10% | 55.87 | 55.81 | 51.63 | 56.78 | 91.4% |
| PLLaVA (Xu et al., 2024) | 10% | 53.89 | 54.17 | 50.89 | 55.85 | 89.2% |
| VisionZip (Yang et al., 2024b) | 10% | 40.78 | 48.42 | 42.56 | 45.75 | 73.6% |
| LLaVA-Scissor (Sun et al., 2025) | 10% | **57.52** | 58.14 | 55.18 | 57.88 | 95.0% |
| ENTROPY-SELECT (Ours) | 10% | 57.21 | **59.80** | **55.56** | **59.30** | **96.3%** |

The initial budget for frame $f$ is $B_f = \lfloor w_f \cdot K \rfloor$. We enforce minimum and maximum constraints per frame: $B_f = \text{clamp}(B_f, B_{\min}, B_{\max})$, where $B_{\min}$ ensures each frame retains at least a minimal representation (typically 4 tokens), and $B_{\max}$ prevents any single frame from dominating the budget.

After applying constraints, we redistribute any remaining tokens to maintain the total budget $K$ by iteratively adding tokens to frames with the highest weights that haven't reached $B_{\max}$.

## 3.5 COMPLEXITY ANALYSIS

Our method maintains linear complexity through efficient local operations and vectorized computation: - Neighbor index construction: $\mathcal{O}(N)$ with caching for repeated grid sizes - Similarity computation: $\mathcal{O}(N \cdot k^2 \cdot D)$ for windows of size $k \times k$ - Entropy calculation: $\mathcal{O}(N \cdot k^2)$ - Gradient computation: $\mathcal{O}(N \cdot D)$ for adjacent differences - Grid-based selection: $\mathcal{O}(g^2 \cdot (N/g^2) \log(N/g^2)) = \mathcal{O}(N \log N)$ - Frame budget allocation: $\mathcal{O}(n)$ for $n$ frames. Since $k$ is a small constant (typically 5-9) and $g$ is fixed (typically 4), the total complexity is $\mathcal{O}(N \cdot D + N \log N)$, which is efficient for practical deployment. The method's complexity remains independent of the compression ratio, providing predictable runtime regardless of how aggressively tokens are compressed. The complete algorithm is detailed in Algorithm 1 (Appendix).

## 4 EXPERIMENTS

### 4.1 EXPERIMENTAL SETUP

**Implementation Details.** We implement ENTROPY-SELECT on the enhanced LLaVA-OneVision model following Sun et al. (2025), which employs SigLIP-SO400M-patch14-384 (Zhai et al., 2023) as the vision encoder and Qwen-2.5-7B (Qwen-Team, 2024) as the language model. Our method operates on extracted visual features without requiring model modifications or access to internal states. Default hyperparameters are set as: spatial window size $w = 5$ for local neighborhoods and entropy-gradient weight $\alpha = 0.5$. All experiments are conducted on a single NVIDIA A100 GPU with 16 frames sampled per video and resolution of $384 \times 384$ pixels per frame.

Table 3: Selected MVBench tasks showing significant improvements by ENTROPY-SELECT. Best results in **bold**.

| Method | RR | AP | OI | MC | MA | FP | CI | Avg. |
|---|---|---|---|---|---|---|---|---|
| LLaVA-OneVision (Li et al., 2024b) | 100% | 52.0 | 46.5 | 67.5 | 53.0 | 77.5 | 79.5 | 62.43 |
| *Token Retention Ratio = 35%* | | | | | | | | |
| DyCoke (Tao et al., 2025) | 35% | 52.5 | 45.0 | 67.0 | 52.5 | 75.5 | 78.5 | 61.78 |
| FastV (Chen et al., 2024b) | 35% | 51.0 | 45.0 | 59.0 | 51.5 | 74.5 | 79.0 | 61.28 |
| PLLaVA (Xu et al., 2024) | 35% | 45.5 | 43.5 | 64.0 | 53.0 | 69.5 | 77.5 | 59.48 |
| VisionZip (Yang et al., 2024b) | 35% | 35.5 | 33.5 | 49.0 | 46.5 | 72.5 | 78.5 | 56.98 |
| LLaVA-Scissor (Sun et al., 2025) | 35% | 54.0 | 49.0 | 60.5 | 52.5 | 77.0 | 79.5 | 61.98 |
| ENTROPY-SELECT (Ours) | 35% | **62.0** | **52.0** | **68.0** | **55.0** | **78.0** | **82.0** | **62.83** |
| *Token Retention Ratio = 10%* | | | | | | | | |
| FastV (Chen et al., 2024b) | 10% | 42.5 | 36.5 | 50.0 | 48.5 | 66.0 | 78.5 | 56.78 |
| PLLaVA (Xu et al., 2024) | 10% | 37.0 | 37.5 | 56.5 | 48.0 | 64.5 | 78.5 | 55.85 |
| VisionZip (Yang et al., 2024b) | 10% | 35.5 | 33.5 | 43.5 | 36.5 | 45.5 | 64.0 | 45.75 |
| LLaVA-Scissor (Sun et al., 2025) | 10% | 48.0 | 41.0 | 51.0 | 48.0 | 76.0 | 80.0 | 57.88 |
| ENTROPY-SELECT (Ours) | 10% | **56.0** | **45.5** | **61.5** | **50.0** | **80.5** | **80.5** | **59.30** |

**Evaluation Benchmarks.** We conduct comprehensive evaluation across three categories of benchmarks: (1) *Video Question-Answering*: ActivityNet-QA (Yu et al., 2019) with 5.8K test samples covering human activities, VideoChatGPT (Maaz et al., 2024) evaluating five aspects of video understanding (correctness, detail, context, temporal, consistency), and NextQA (Xiao et al., 2021) focusing on causal and temporal reasoning; (2) *Long Video Understanding*: EgoSchema (Mangalam et al., 2023) with 5K egocentric videos averaging 180 seconds, MLVU (Zhou et al., 2024b) spanning 3 minutes to 2 hours, and VideoMME (Fu et al., 2024a) with diverse domains; (3) *Comprehensive Tasks*: MVBench (Li et al., 2024d) comprising 20 diverse video understanding tasks in multiple-choice format. Comparative results are taken from Sun et al. (2025), with our method following the same evaluation protocol.

## 4.2 MAIN RESULTS

**Long Video Understanding and Comprehensive Tasks.** Table 2 reports results at 50%, 35%, and 10% token retention. At 50%, performance across methods remains close to the full-token baseline; our method achieves the highest average (99.9%) and leads on MLVU, VideoMME, and MVBench, while FastV is strongest on EgoSchema. At 35%, our method attains the top average (99.8%) and ranks first on all four benchmarks; notably, on EgoSchema our compressed models at 50% and 35% retention even surpass the full-token baseline, indicating beneficial denoising/regularization from entropy-guided selection. At 10% retention, our method maintains the best average (96.3%), leading on MLVU, VideoMME, and MVBench, and remaining competitive on EgoSchema where LLaVA-Scissor is best. Overall, local-entropy selection preserves salient information, can enhance performance on EgoSchema at moderate compression, and degrades gracefully as retention decreases.

**Video Question-Answering Performance.** Table 4 presents a comprehensive comparison on ActivityNet-QA, VideoChatGPT with five aspect scores and the averaged score, and NextQA across three retention ratios. At moderate compression of 50 percent, most methods remain close to the full-token baseline LLaVA-OneVision (Li et al., 2024b). LLaVA-Scissor (Sun et al., 2025), FastV (Chen et al., 2024b), DyCoke (Tao et al., 2025), and PLLaVA (Xu et al., 2024) form a tight cluster, while the proposed ENTROPY-SELECT attains the highest normalized average at 99.73 percent, showing that information-theoretic selection can match strong training-free baselines without access to model internals. At 35 percent retention, performance dispersion increases. ENTROPY-SELECT achieves the best ActivityNet accuracy at 48.34 and ties for the top CO score, resulting in the highest normalized average at 99.84 percent. LLaVA-Scissor remains competitive, whereas attention- or similarity-driven baselines such as VisionZip (Yang et al., 2024b) exhibit more pronounced declines.

Under aggressive compression of 10 percent, robustness differences become salient. LLaVA-Scissor (Sun et al., 2025) yields the strongest normalized average at 98.62 percent and maintains solid scores across ActivityNet and VideoChatGPT aspects, reflecting the utility of connectivity-aware selection

Table 4: Performance comparison on video question-answering benchmarks. Only Avg.(%) is highlighted: best in **bold**, second-best underlined.

| Method | RR | ActivityNet | | VideoChatGPT | | | | | | NextQA | Avg.(%) |
|---|---|---|---|---|---|---|---|---|---|---|---|
| | | Acc. | Score | CI | DO | CU | TU | CO | Avg | | |
| LLaVA-OneVision (Li et al., 2024b) | 100% | 48.09 | 3.47 | 3.37 | 3.78 | 3.52 | 3.02 | 2.63 | 3.26 | 81.33 | 100% |
| *Token Retention Ratio = 50%* | | | | | | | | | | | |
| FastV (Chen et al., 2024b) | 50% | 47.95 | 3.47 | 3.36 | 3.77 | 3.50 | 2.99 | 2.57 | 3.24 | 81.11 | 99.47% |
| DyCoke (Tao et al., 2025) | 50% | 47.88 | 3.47 | 3.33 | 3.76 | 3.51 | 3.01 | 2.58 | 3.24 | 81.06 | 99.41% |
| PLLaVA (Xu et al., 2024) | 50% | 47.59 | 3.45 | 3.36 | 3.73 | 3.52 | 3.00 | 2.66 | 3.25 | 81.04 | 99.33% |
| VisionZip (Yang et al., 2024b) | 50% | 45.42 | 3.47 | 3.16 | 3.63 | 3.34 | 2.75 | 2.61 | 3.10 | 78.46 | 96.59% |
| LLaVA-Scissor (Sun et al., 2025) | 50% | 47.89 | 3.47 | 3.37 | 3.76 | 3.47 | 3.00 | 2.65 | 3.25 | 81.12 | 99.45% |
| ENTROPY-SELECT (Ours) | 50% | 48.09 | 3.47 | 3.50 | 3.04 | 3.79 | 2.55 | 3.35 | 3.25 | 81.41 | **99.73%** |
| *Token Retention Ratio = 35%* | | | | | | | | | | | |
| FastV (Chen et al., 2024b) | 35% | 47.83 | 3.46 | 3.32 | 3.74 | 3.47 | 2.97 | 2.61 | 3.22 | 80.49 | 98.99% |
| DyCoke (Tao et al., 2025) | 35% | 47.81 | 3.45 | 3.31 | 3.74 | 3.46 | 2.98 | 2.54 | 3.21 | 80.86 | 99.17% |
| PLLaVA (Xu et al., 2024) | 35% | 47.23 | 3.42 | 3.26 | 3.70 | 3.39 | 2.92 | 2.59 | 3.17 | 79.66 | 98.24% |
| VisionZip (Yang et al., 2024b) | 35% | 44.69 | 3.46 | 3.13 | 3.61 | 3.31 | 2.71 | 2.57 | 3.07 | 77.72 | 95.86% |
| LLaVA-Scissor (Sun et al., 2025) | 35% | 47.88 | 3.48 | 3.03 | 3.76 | 2.61 | 3.35 | 2.63 | 3.25 | 80.83 | 99.33% |
| ENTROPY-SELECT (Ours) | 35% | 48.34 | 3.47 | 3.48 | 3.03 | 3.76 | 2.61 | 3.35 | 3.25 | 81.25 | **99.84%** |
| *Token Retention Ratio = 10%* | | | | | | | | | | | |
| FastV (Chen et al., 2024b) | 10% | 44.95 | 3.38 | 3.04 | 3.60 | 3.28 | 2.80 | 2.49 | 3.04 | 78.76 | 96.36% |
| PLLaVA (Xu et al., 2024) | 10% | 45.28 | 3.37 | 3.11 | 3.56 | 3.25 | 2.78 | 2.55 | 3.05 | 77.87 | 96.22% |
| VisionZip (Yang et al., 2024b) | 10% | 38.58 | 3.30 | 2.65 | 3.09 | 2.73 | 2.31 | 2.42 | 2.64 | 65.09 | 82.35% |
| LLaVA-Scissor (Sun et al., 2025) | 10% | 47.75 | 3.46 | 3.26 | 3.68 | 3.41 | 2.90 | 2.52 | 3.15 | 80.03 | **98.62%** |
| ENTROPY-SELECT (Ours) | 10% | 46.76 | 3.44 | 3.35 | 2.90 | 3.65 | 2.51 | 3.25 | 3.13 | 80.52 | 97.20% |

in extreme token budgets. ENTROPY-SELECT remains competitive at 97.20 percent, preserving balance across CI, DO, CU, TU, and CO, and securing a strong NextQA accuracy of 80.52. It surpasses FastV (Chen et al., 2024b) and PLLaVA (Xu et al., 2024) and substantially exceeds VisionZip (Yang et al., 2024b), which shows the largest degradation. Across aspects, temporal understanding exhibits the steepest decline for weaker approaches at low retention, whereas methods emphasizing semantic structure or information-theoretic diversity maintain temporal cues more effectively. These results indicate that entropy-guided selection delivers state-of-the-art or near–state-of-the-art accuracy at 35 to 50 percent retention and remains resilient at 10 percent, supporting its value as a training-free and architecture-agnostic strategy.

**MVBench Multi-Task Performance.** Table 3 highlights representative MVBench tasks where ENTROPY-SELECT shows notable gains, and Figure 3 provides a 20-task radar view at 35% retention. Notably, with only 35% tokens retained our method attains an average of 62.83 on MVBench, slightly exceeding the full-token baseline (62.43), indicating an "enhancement under compression." We hypothesize three contributing factors: (i) redundancy trimming increases the signal-to-noise ratio by suppressing homogeneous background tokens; (ii) entropy–gradient fusion prioritizes boundary/structure tokens that are disproportionately informative for recognition; and (iii) grid-based coverage prevents over-concentration, yielding a more balanced spatial summary that benefits downstream reasoning.

Gains concentrate on fine-grained and spatiotemporal tasks. For Action Prediction (AP), we reach 62.0 at 35% retention, compared to 54.0 reported by LLaVA-Scissor (Sun et al., 2025), suggesting better preservation of motion cues. In Moving Count (MC), our 68.0 surpasses all baselines, reflecting stronger maintenance of object trajectories. Counterfactual Inference (CI) shows one of the largest margins (82.0), implying that local-entropy selection helps retain causal visual evidence while reducing distractors. Complete results across all 20 MVBench tasks are provided in Appendix A.1.

## 4.3 ABLATION STUDIES

We conduct ablations to quantify the contribution of each component in ENTROPY-SELECT and to study how hyperparameters affect performance.

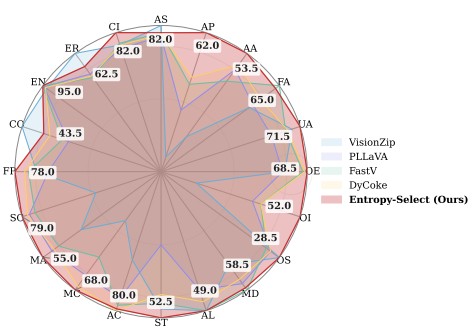

Figure 3: MVBench 35% retention radar.

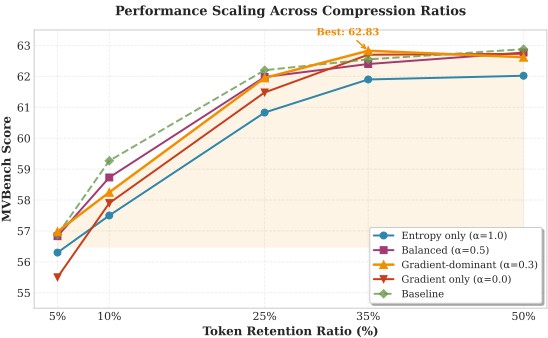

Figure 4: Retention scaling under gradient-dominant fusion.

**Entropy vs. Gradient Weight.** Table 5 evaluates the impact of varying entropy ($\alpha$) and gradient (importance) weights in the combined score $S_i$ at 35% retention ratio. Pure entropy ($\alpha = 1.0$, gradient=0.0) provides strong baseline performance by prioritizing information diversity. Entropy-dominant fusion ($\alpha = 0.7$, gradient=0.3) enhances structural preservation, improving scores (e.g., +0.67 on VideoMME over pure entropy). Gradient-dominant fusion ($\alpha = 0.3$, gradient=0.7) focuses more on edges but slightly reduces temporal robustness. Pure gradient ($\alpha = 0.0$, gradient=1.0) shows the lowest performance, confirming entropy's critical role.

Table 5: Results of entropy vs. gradient weights on MVBench.

| Entropy Weight ($\alpha$) | Gradient Weight | MVBench Score at Different Retention Ratios | | | | |
|---|---|---|---|---|---|---|
| | | 5% | 10% | 25% | 35% | 50% |
| 1.0 (entropy only) | 0.0 | 56.30 | 57.50 | 60.83 | 61.90 | 62.02 |
| 0.7 | 0.3 | 55.60 | 58.42 | 62.12 | 61.95 | 62.70 |
| 0.5 | 0.5 | 56.83 | 58.73 | 61.98 | 62.40 | 62.77 |
| 0.3 | 0.7 | **56.98** | **58.25** | **61.95** | **62.83** | **62.62** |
| 0.0 (gradient only) | 1.0 | 55.50 | 57.90 | 61.48 | 62.70 | 62.73 |

**Window Size and Compression Scaling.** Table 8 studies spatial/temporal window sizes, and Figure 4 visualizes their impact across retention ratios. A moderate spatial window (SW=5) and temporal window (TW=3) provide the best balance: smaller SW slightly helps at higher retention due to sharper localization, whereas larger SW dilutes local patterns; TW=3 consistently improves robustness by capturing cross-frame dependencies. Under the gradient-dominant fusion ($\alpha = 0.3$), performance remains stable as retention decreases (only about 9% drop from 50% to 5%), showing that the same configuration supports both fine-detail preservation at higher retention and graceful degradation at aggressive compression.

**Temperature Sensitivity.** We ablate the softmax temperature $\tau$ used in computing attention-based similarity distributions. As shown in Table 6, performance remains stable across a wide range of $\tau \in \{0.25, 0.5, 1.0, 1.25, 1.5, 2.0, 5.0\}$, with the maximum difference being only 0.45 points on MVBench at 35% retention. This robustness stems from the normalized entropy formulation $\hat{H}_i = H(p_{ij})/\log(|\mathcal{N}_i| - 1)$, which preserves relative token rankings even under substantial temperature variations.

Table 6: Ablation on temperature $\tau$ (left) and grid size $g$ (right) on MVBench at 35% retention.

| $\tau$ | 0.25 | 0.5 | 1.0 | 1.25 | 1.5 | 2.0 | 5.0 | $g$ | 3 | 4 | 5 |
|---|---|---|---|---|---|---|---|---|---|---|---|
| Score | 62.38 | 62.45 | 62.83 | 62.45 | 62.62 | 62.62 | 62.58 | Score | 62.55 | 62.83 | 62.58 |

**Grid Size for Coverage Enforcement.** We also ablate the grid partition size $g$ used for spatial coverage constraints. Table 6 shows that performance varies by at most 0.28 points across $g \in$

{3, 4, 5}, indicating that the coverage mechanism is robust as long as the grid granularity remains coarse.

**Cross-Architecture Generalization.** To validate architecture-agnosticism, we evaluate ENTROPY-SELECT on Qwen3-VL-8B (Bai et al., 2025) and InternVL3-9B (Zhu et al., 2025), which differ substantially from LLaVA-OneVision: Qwen3-VL uses 2D-RoPE for dynamic resolution, while InternVL3 employs pixel unshuffle to reduce tokens to 1/4 count (256 per 448×448 tile). As shown in Table 7, ENTROPY-SELECT achieves consistent performance across all three models using *identical default hyperparameters*, confirming generalization across diverse vision encoders and tokenization strategies.

Table 7: Cross-architecture evaluation on MVBench at 35% retention. All models use identical hyperparameters without any architecture-specific tuning.

| Model | Vision Encoder | Original | Ours (35%) | Rel. Perf. |
|---|---|---|---|---|
| LLaVA-OneVision (Li et al., 2024b) | SigLIP-SO400M | 62.43 | 62.83 | 100.6% |
| Qwen3-VL-8B (Bai et al., 2025) | Qwen3-ViT | 62.15 | 62.41 | 100.4% |
| InternVL3-9B (Zhu et al., 2025) | InternViT | 72.18 | 71.95 | 99.7% |

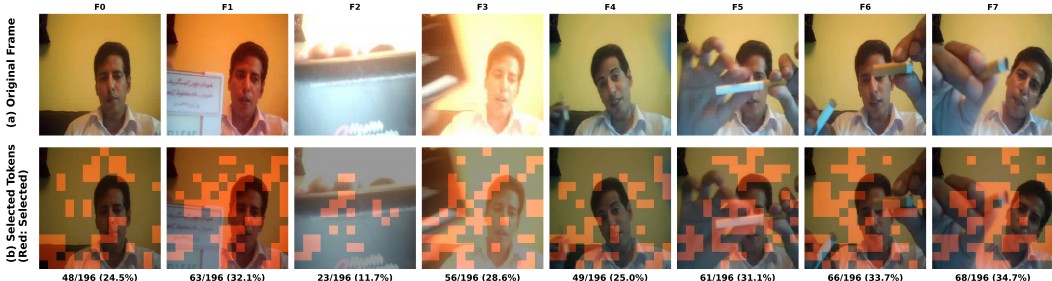

Figure 5: Token selection visualization at 30% retention.

**Qualitative Analysis.** Figure 5 visualizes token selection across consecutive video frames at 30% retention. The selection pattern reveals adaptive frame-level allocation: transitional or blurry frames (F2) receive minimal tokens, while action-rich frames with hand-object interactions (F4-F7) receive denser coverage. Within each frame, selections concentrate on human silhouettes, facial regions, and manipulation zones while suppressing homogeneous backgrounds, confirming that entropy-guided pruning identifies semantically relevant content.

## 4.4 COMPUTATIONAL EFFICIENCY ANALYSIS

All latency measurements in this subsection are obtained on 16-frame videos. Under this setting, our method attains predictable, ratio-agnostic selection cost: as shown in Fig. 1a, ENTROPY-SELECT maintains a nearly flat processing time across a wide range of compression ratios, in stark contrast to LLaVA-Scissor whose latency increases as compression tightens. This behavior matches the algorithmic design: fixed-size windows, grid quotas, and a single global sort yield an $\mathcal{O}(N \log N)$ selector whose runtime is effectively decoupled from the user-chosen retention. The stability across settings enables capacity planning without tuning overhead and removes the typical accuracy–latency trade-off induced by iterative or graph-based pruning.

Beyond selector cost, our end-to-end breakdown at a representative 35% retention on 16-frame inputs (Fig. 1b) shows where time is spent in practice: encoder time remains unchanged ( 25-26 ms), while token mapping decreases from 485 ms to 445 ms and LLM prefill drops from 101 ms to 38 ms for ENTROPY-SELECT (compared to 703 ms mapping and 41 ms prefill for LLaVA-Scissor), yielding totals of 508 ms (ENTROPY-SELECT) versus 611 ms (baseline) versus 770 ms (Scissor). Thus, ENTROPY-SELECT reduces the token-mapping stage while keeping the vision encoder and LLM forward pass unchanged, delivering lower total latency than both the LLaVA-Scissor pipeline and the baseline without token compression. When projector overhead dominates, fewer tokens

directly reduce projector and attention time, making compressed inference strictly faster once the selector's small fixed overhead is amortized.

Table 8: Impact of window sizes on MVBench. SW: Spatial Window, TW: Temporal Window, TSW: Temporal Spatial Window.

| Configuration | SW | TW | TSW | MVBench Score at Different RR | | | |
|---|---|---|---|---|---|---|---|
| | | | | 10% | 25% | 35% | 50% |
| Baseline | 5 | 3 | 3 | 59.27 | 62.20 | 62.55 | 62.88 |
| *Spatial Window Variations* | | | | | | | |
| SW=3 | 3 | 3 | 3 | 59.40 | 62.42 | 62.58 | 63.10 |
| SW=7 | 7 | 3 | 3 | 58.50 | 62.02 | 62.38 | 62.88 |
| SW=9 | 9 | 3 | 3 | 59.05 | 61.62 | 62.73 | 63.08 |
| *Temporal Window Variations* | | | | | | | |
| No TW | 5 | 1 | - | 59.20 | 61.62 | 62.08 | 62.92 |
| TW=2 | 5 | 2 | 3 | 59.08 | 62.23 | 62.62 | 62.85 |

## 5 DISCUSSION

Our results suggest that information-theoretic selection offers a stable and portable alternative to heuristic saliency proxies. By prioritizing tokens with high local neighborhood entropy and reinforcing structural cues via gradients–while enforcing spatial coverage–ENTROPY-SELECT achieves near-baseline accuracy at moderate retention and graceful degradation at 10%. Notably, on EgoSchema we observe "enhancement under compression" at both 50% and 35% retention, indicating that trimming low-entropy background can denoise inputs and emphasize boundary/detail tokens that matter for egocentric scenes. A plausible hypothesis is that entropy-guided pruning suppresses low-information, correlated background tokens and thus reduces cross-token interference in the projector/attention stack, effectively acting as a denoising prior that sharpens boundaries and salient motion cues.

Limitations include mixed gains on MVBench narrative/order-sensitive tasks (e.g., CO/ER) under aggressive compression, sensitivity of entropy estimation to representation quality in small-object regimes, and the need for broader community validation across heterogeneous model architectures (e.g., varied visual backbones and vision–language alignment modules) to establish portability. Future work will explore adaptive multi-scale neighborhoods, lightweight temporal anchors for global cues, task-adaptive fusion/coverage policies that remain training-free and architecture-agnostic, and further reducing selector overhead via parallel execution and principled approximations.

## 6 CONCLUSION

We presented ENTROPY-SELECT, a training-free, architecture-agnostic token selector that scores tokens by local neighborhood entropy, fuses structural gradients, and enforces spatial coverage with saliency-weighted frame budgets, yielding an $\mathcal{O}(N \log N)$ selection whose runtime is effectively decoupled from the retention ratio. Operating purely on exported features, it is plug-and-play for black-box VLMs and, across MVBench, ActivityNet-QA, VideoMME, and EgoSchema, matches or surpasses full-token baselines at moderate retention while degrading gracefully under aggressive compression; notably, we observe enhancement under compression on MVBench and EgoSchema at 35–50% retention, suggesting that entropy-guided pruning can denoise inputs by eliminating low-information redundancy and emphasizing semantically rich boundaries. From a deployment perspective, the selector's small, predictable overhead is amortized by reduced projector and LLM attention costs, making compressed inference strictly faster than no compression while enabling scalable processing of long-duration videos in resource-constrained environments. Future work will pursue adaptive multi-scale neighborhoods, lightweight temporal anchors for narrative cues, and system-level speedups via GPU-parallel entropy kernels, batched/partial sorting, and overlap with vision encoding, alongside explorations of task-adaptive entropy thresholds for specialized domains.

## REPRODUCIBILITY STATEMENT

We will release full training and inference code and all pretrained checkpoints upon publication, together with a pinned environment file. All hyperparameters (e.g., retention ratios, fusion coefficients, batch sizes, learning rates, decoding settings) are fully disclosed in the paper and configs, and precise pseudocode for the token compression and fusion algorithms is provided in the supplementary. We use only public benchmarks with official evaluation protocols and document data acquisition and preprocessing (frame sampling, resolution, tokenization) for exact replication. One-command scripts with fixed random seeds reproduce the main tables and figures, and we document any residual nondeterminism with deterministic execution flags.

## ETHICS STATEMENT

This work aims to improve computational efficiency of video understanding systems, potentially democratizing access to advanced AI capabilities. We acknowledge potential dual-use concerns in surveillance applications and recommend deployment guidelines prioritizing privacy protection. All datasets used are publicly available with proper licenses. No human subjects were involved in this research.

## LLM USAGE DISCLOSURE

We used Claude-4 for grammar checking and improving paper clarity in Sections 1 and 6. All technical content, experimental design, and analysis are original human work. LLM suggestions were manually reviewed and modified to ensure accuracy.

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

## A APPENDIX

### A.1 COMPLETE MVBENCH RESULTS

Abbreviations: AS=Action Sequence, AP=Action Prediction, AA=Action Antonym, FA=Fine-grained Action, UA=Unexpected Action, OE=Object Existence, OI=Object Interaction, OS=Object Shuffle, MD=Moving Direction, AL=Action Localization, ST=Scene Transition, AC=Action Count, MC=Moving Count, MA=Moving Attribute, SC=State Change, FP=Fine-grained Pose, CO=Character Order, EN=Egocentric Navigation, ER=Episodic Reasoning, CI=Counterfactual Inference.

Table 9 provides a granular view of where our method helps most. At 35% retention, Ours attains the highest average (62.83 vs. 61.78 DyCoke and 61.28 FastV), with clear wins on AP (62.0; +9.5 over next-best 52.5), OI (52.0; +7.0), CI (82.0; +3.0), FP (78.0; +2.5), AA (53.5; +2.5), UA (71.5; +2.0), MA (55.0; +2.0), MC (68.0; +1.0), ST (52.5; +1.5), SC (79.0; +1.5), and OE (68.5; +0.5). We tie for best on AL (49.0; with FastV and VisionZip) and EN (95.0; with FastV), are close on MD (58.5; -0.5 to FastV), AC (80.0; -0.5 to DyCoke), FA (65.0; -1.0 to FastV), and AS (82.0; -2.0 to VisionZip), while trailing on sequence/order and longer-range narrative tasks CO (43.5; -3.5 to VisionZip) and ER (62.5; -3.5 to VisionZip). At 10% retention, Ours maintains the best average (59.30 vs. 57.88 LLaVA-S) and achieves top scores on AP, AA, UA, OI, ST, AC, MC, MA, SC, FP, CO, and CI;

Table 9: Complete MVBench results across all 20 tasks (transposed for readability).

| Task | 35% Retention | | | | | 10% Retention | | | | |
|------|--------|-------|--------|-----------|------|-------|--------|-----------|---------|------|
| | DyCoke | FastV | PLLaVA | VisionZip | **Ours** | FastV | PLLaVA | VisionZip | LLaVA-S | **Ours** |
| AS | 81.5 | 82.0 | 81.0 | 84.0 | **82.0** | 78.5 | 77.5 | 75.5 | 80.5 | 79.5 |
| AP | 52.5 | 51.0 | 45.5 | 35.5 | **62.0** | 42.5 | 37.0 | 35.5 | 48.0 | **56.0** |
| AA | 51.0 | 47.5 | 50.0 | 35.5 | **53.5** | 37.0 | 42.5 | 27.5 | 47.5 | **50.0** |
| FA | 61.0 | 66.0 | 59.0 | 58.0 | 65.0 | 67.0 | 54.0 | 42.0 | 59.0 | 61.5 |
| UA | 68.5 | 68.0 | 64.5 | 69.5 | **71.5** | 63.0 | 63.0 | 47.0 | 66.5 | **67.5** |
| OE | 68.0 | 67.5 | 63.0 | 63.5 | **68.5** | 67.0 | 60.0 | 48.0 | 64.5 | 63.5 |
| OI | 45.0 | 45.0 | 43.5 | 33.5 | **52.0** | 36.5 | 37.5 | 33.5 | 41.0 | **45.5** |
| OS | 27.0 | 27.0 | 27.5 | 28.5 | 28.5 | 27.0 | 28.0 | 29.0 | 28.0 | 28.0 |
| MD | 56.5 | 59.0 | 57.5 | 53.0 | 58.5 | 55.5 | 51.5 | 49.5 | 56.5 | 57.5 |
| AL | 47.5 | 49.0 | 45.5 | 49.0 | 49.0 | 46.0 | 46.0 | 39.0 | 45.0 | 48.0 |
| ST | 48.5 | 50.0 | 40.0 | 51.0 | 52.5 | 47.5 | 35.0 | 33.0 | 43.0 | **50.0** |
| AC | 80.5 | 79.0 | 77.0 | 66.5 | 80.0 | 65.5 | 70.5 | 46.5 | 63.0 | **72.0** |
| MC | 67.0 | 59.0 | 64.0 | 49.0 | **68.0** | 50.0 | 56.5 | 43.5 | 51.0 | **61.5** |
| MA | 52.5 | 51.5 | 53.0 | 46.5 | **55.0** | 48.5 | 48.0 | 36.5 | 48.0 | **50.0** |
| SC | 77.5 | 75.5 | 77.0 | 59.0 | **79.0** | 65.0 | 73.5 | 46.0 | 64.5 | **74.0** |
| FP | 75.5 | 74.5 | 69.5 | 72.5 | **78.0** | 66.0 | 64.5 | 45.5 | 76.0 | **80.5** |
| CO | 42.5 | 40.5 | 39.5 | 47.0 | 43.5 | 40.0 | 39.5 | 37.5 | 39.5 | 42.0 |
| EN | 94.0 | 95.0 | 94.5 | 93.5 | 95.0 | 93.5 | 93.5 | 81.5 | 95.5 | 94.0 |
| ER | 60.0 | 59.5 | 60.5 | 66.0 | 62.5 | 61.0 | 60.5 | 54.5 | 60.5 | 60.5 |
| CI | 78.5 | 79.0 | 77.5 | 78.5 | **82.0** | 78.5 | 78.5 | 64.0 | 80.0 | 80.5 |
| Avg. | 61.78 | 61.28 | 59.48 | 56.98 | **62.83** | 56.78 | 55.85 | 45.75 | 57.88 | **59.30** |

we are also best on MD (57.5) and AL (48.0), tie on ER (60.5) with several methods but slightly below FastV (61.0), and remain close on AS (79.5; -1.0 to LLaVA-S), OE (63.5; -1.0 to LLaVA-S), and EN (94.0; -1.5 to LLaVA-S). The pattern is consistent with our design: entropy-plus-gradient scoring concentrates budget on action-relevant regions and boundaries, which benefits fine-grained, counting, attribute, and causal reasoning tasks, while grid-based coverage preserves enough global context to remain competitive on scene-level reasoning even under aggressive compression.

## A.2 COMPUTATIONAL EFFICIENCY DETAILS

Table 10: Detailed computational efficiency metrics across compression ratios.

| Base Rate | Original Tokens | Compressed Tokens | Time (ms) | Reduction (%) |
|-----------|-----------------|-------------------|-----------|---------------|
| 0.05 | 8,467,200 | 422,550 | 440.973 | 95.01 |
| 0.10 | 8,467,200 | 846,450 | 454.652 | 90.00 |
| 0.35 | 8,467,200 | 2,963,250 | 461.727 | 65.00 |
| 0.50 | 8,467,200 | 4,233,600 | 465.074 | 50.00 |

As shown in Table 10, compressed tokens scale near-linearly with the base rate (0.05 to 0.50: 0.42M to 4.23M), while end-to-end latency remains almost flat (441 to 465 ms, 5% increase), consistent with a latency model $T \approx T_0 + k \cdot N_{ret}$ where fixed costs dominate; small deviations between target and realized token counts arise from grid/window rounding at boundaries.

## A.3 ALGORITHM DESCRIPTION

Algorithm 1 selects informative tokens in four fixed-cost steps without model retraining or internal-state access. First, for each token we compute temperature-scaled local-window similarities and the normalized Shannon entropy to quantify neighborhood unpredictability (information diversity). Second, we extract L2 gradient magnitudes on the token grid to capture structural edges; entropy and gradients are linearly fused into a single importance score. Third, for videos we allocate a per-frame token budget via a softmax over robust frame saliency (top-q mean), enforcing simple min/max caps. Finally, within each frame we ensure spatial coverage by selecting a quota per grid cell and then fill remaining budget by global top scores; optional grid fusion summarizes discarded tokens by score-weighted averaging. All windows/grids are fixed, and selection reduces to one global sort,

---

**Algorithm 1** ENTROPY-SELECT: Information-Theoretic Token Compression

---

1: **Input:** Video tokens $\mathbf{T} \in \mathbb{R}^{N \times D}$, frames $F$, spatial dimension $S$, compression rate $r$
2: **Output:** Compressed tokens $\mathbf{T}' \in \mathbb{R}^{K \times D}$ where $K = \lfloor r \cdot N \rfloor$
3:
4: **// Phase 1: Local Entropy Computation**
5: **for** each token $\mathbf{t}_i$ in $\mathbf{T}$ **do**
6:    Normalize: $\hat{\mathbf{t}}_i \leftarrow \mathbf{t}_i / \|\mathbf{t}_i\|_2$
7:    Get spatial neighbors within window size $w$: $\mathcal{N}_i$
8:    Compute similarities: $s_{ij} = \hat{\mathbf{t}}_i^T \hat{\mathbf{t}}_j$ for $j \in \mathcal{N}_i$
9:    Apply temperature softmax: $p_{ij} = \exp(s_{ij}/\tau) / \sum_j \exp(s_{ij}/\tau)$
10:   Calculate entropy: $H_i = -\sum_j p_{ij} \log p_{ij}$
11:   Normalize: $\hat{H}_i = H_i / \log(|\mathcal{N}_i| - 1)$
12: **end for**
13:
14: **// Phase 2: Gradient Computation**
15: **for** each spatial position $(x, y)$ **do**
16:   Compute gradients: $G_{x,y} = \sqrt{\|\nabla_x \mathbf{t}_{x,y}\|^2 + \|\nabla_y \mathbf{t}_{x,y}\|^2}$
17:   Normalize: $\hat{G}_{x,y} = (G_{x,y} - G_{min})/(G_{max} - G_{min})$
18: **end for**
19:
20: **// Phase 3: Score Fusion and Frame Budget Allocation**
21: Combine scores: $S_i = \alpha \cdot \hat{H}_i + (1 - \alpha) \cdot \hat{G}_i$
22: **if** multiple frames **then**
23:   **for** each frame $f$ **do**
24:     Compute robust mean score using top-$q$ tokens
25:   **end for**
26:   Allocate budgets via softmax: $B_f = \text{softmax}(\bar{S}_f / \tau_{alloc}) \cdot K$
27: **else**
28:   $B = K$ for single frame
29: **end if**
30:
31: **// Phase 4: Spatial-Aware Selection**
32: **for** each frame $f$ **do**
33:   Divide into $g \times g$ spatial grids
34:   **for** each grid region $R$ **do**
35:     Select top-$k_R$ tokens where $k_R = \max(1, \lfloor B_f \cdot \rho/g^2 \rfloor)$
36:   **end for**
37:   Select remaining budget from global top scores
38:   **if** grid fusion enabled **then**
39:     Compute weighted average of unselected tokens per grid
40:     Anchor fused tokens to highest-scoring unselected positions
41:   **end if**
42: **end for**
43: **return** Selected and fused tokens $\mathbf{T}'$

---

yielding overall complexity $\mathcal{O}(N, w^2 D + N \log N)$ with runtime effectively decoupled from the retention ratio.

## A.4 VISUALIZATION RESULTS

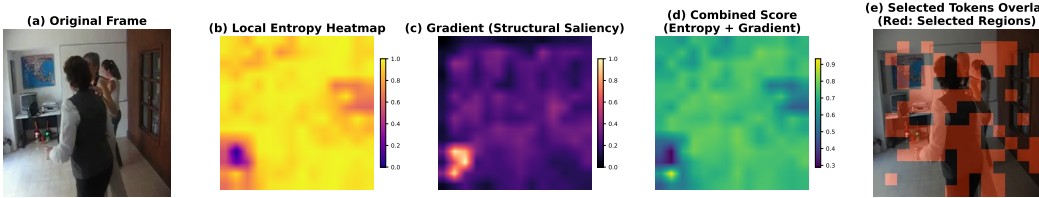

Figure 6: Visualization of ENTROPY-SELECT token compression at 35% retention ratio. (a) Original video frame showing two people in an indoor scene. (b) Local entropy heatmap where higher values indicate tokens in heterogeneous neighborhoods. (c) Gradient-based structural saliency highlighting edges and boundaries. (d) Combined score integrating entropy and gradient components. (e) Selected tokens overlay demonstrating preservation of semantically important regions.

Figure 6 illustrates how ENTROPY-SELECT prioritizes informative regions. The local-entropy map (b) highlights heterogeneous neighborhoods around people and object boundaries, while the gradient saliency (c) emphasizes edges and fine structures. Their fusion (d) produces a balanced importance map that concentrates on subjects, interaction zones, and high-frequency transitions, yet maintains coverage of secondary context. The final selections (e) show dense retention on human silhouettes and manipulable objects, with sparsity in homogeneous background areas—consistent with our hypothesis that removing low-entropy background denoises inputs. This selection pattern explains the observed gains on fine-grained and spatiotemporal tasks and the "enhancement under compression" on EgoSchema at 50% and 35% retention.

