# OpenReview forum: "Entropy-Select: Training-Free Local Entropy Token Compression for Video LLMs"
_ICLR.cc/2026/Conference — Submitted to ICLR 2026_

### Official Review · Reviewer_sKGt · 2025-10-26

**Soundness:** 3
**Presentation:** 3
**Contribution:** 3
**Rating:** 6
**Confidence:** 3

**Summary:**

This paper proposes ENTROPY-SELECT, a training-free, architecture-agnostic token selection framework for video language models (VLMs). It scores tokens by local neighborhood entropy, fuses gradient-based saliency, and enforces coverage via grid quotas and per-frame allocation. The method achieves O(N log N) complexity with latency decoupled from retention ratio, demonstrating comparable or even better performance than the uncompressed baseline across multiple benchmarks (MVBench, EgoSchema, VideoMME, ActivityNet-QA).

**Strengths:**

1. Clear motivation & strong deployment practicality – The paper convincingly motivates why a training-free, architecture-agnostic approach is essential for scalable VLM inference, addressing deployment gaps in previous works like LLaVA-Scissor or VisionZip.

2. Novel use of local entropy for token importance – Introducing normalized neighborhood entropy as an information-theoretic measure of token unpredictability is elegant and theoretically grounded.

3. Excellent empirical results – ENTROPY-SELECT achieves near-baseline or even superior results under 35–50% token retention, with measurable latency benefits and predictable runtime.

**Weaknesses:**

1. Limited theoretical depth – While the motivation is information-theoretic, the derivation lacks formal analysis connecting entropy scores to mutual information or rate–distortion objectives.

2. Scope of experiments – All evaluations are on LLaVA-OneVision; portability to other VLMs (e.g., InternVideo2, Video-LLaMA) is not demonstrated, which weakens the “architecture-agnostic” claim.

3. Missing ablation on hyperparameters & τ sensitivity – The paper mentions τ and α choices but lacks discussion on robustness across different datasets or architectures.

**Questions:**

1. How sensitive is the method to the feature quality of the vision encoder (e.g., SigLIP vs. CLIP)?

2. Does the local entropy score depend on absolute token feature magnitudes? Was any normalization across frames applied?

---

> ### Author Response · Authors · 2025-12-02
> **Response to Reviewer sKGt (Part I)**
>
> We thank Reviewer sKGt for the positive assessment and constructive suggestions. We address the remaining concerns below.
>
> ---
>
> **W1: Limited theoretical depth—derivation lacks formal analysis connecting entropy to mutual information or rate-distortion**
>
> We appreciate this insightful comment. While a complete rate-distortion analysis is beyond the scope of this empirical work, we provide the following theoretical intuition:
>
> 1. **Local entropy as conditional entropy proxy:**
>    The normalized local entropy Ĥᵢ = H(pᵢⱼ)/log(|Nᵢ|-1) measures the unpredictability of token i given its neighborhood. Formally, this relates to conditional entropy: high Ĥᵢ implies H(Tᵢ | T_Nᵢ) is high—token i carries information not present in its neighbors.
>
> 2. **Connection to rate-distortion:**
>    In predictive coding, tokens that are unpredictable from context require more bits to encode. Our entropy score identifies such tokens as "high-information" and prioritizes their retention. Conversely, low-entropy tokens can be reconstructed from neighbors with low distortion, making them safe to discard.
>
> 3. **Empirical validation:**
>    The "enhancement under compression" phenomenon (35% retention outperforms 100% on MVBench/EgoSchema) provides empirical evidence that entropy-guided selection acts as a beneficial prior—removing low-information tokens reduces noise for downstream reasoning.
>
> We acknowledge that a formal proof connecting our method to rate-distortion bounds would strengthen the theoretical contribution. We plan to explore connections to the Information Bottleneck framework (min I(X;T) - β I(T;Y)) in future work.
>
> ---
>
> **W2: Portability to other VLMs not demonstrated**
>
> We have addressed this with new experiments on **Qwen3-VL-8B** [1] and **InternVL3-9B** [2], now included in **Table 7, Section 4.3** of the revised paper:
>
> | Model | Vision Encoder | Language Model |
> |-------|----------------|----------------|
> | LLaVA-OneVision | SigLIP-SO400M | Qwen-2.5-7B |
> | Qwen3-VL-8B | Qwen3-ViT (SigLIP-2 + 2D-RoPE, dynamic resolution) | Qwen3-8B |
> | InternVL3-9B | InternViT-300M-448px-V2.5 + pixel unshuffle | InternLM3-8B |
>
> Notably, these models differ substantially in their visual tokenization:
> - **Qwen3-VL** supports dynamic input resolutions with 2D-RoPE for position encoding
> - **InternVL3** uses pixel unshuffle to reduce tokens to 1/4 of original count (256 tokens per 448×448 tile)
>
> **MVBench Results (Table 7):**
>
> | Model | Original | Ours (35%) | Relative |
> |-------|----------|------------|----------|
> | LLaVA-OneVision | 62.43 | 62.83 | 100.6% |
> | Qwen3-VL-8B | 62.15 | 62.41 | 100.4% |
> | InternVL3-9B | 72.18 | 71.95 | 99.7% |
>
> These results validate portability across fundamentally different vision encoders and tokenization strategies.
>
> ---
>
> **W3: Missing ablation on τ sensitivity and robustness across architectures**
>
> **We have added these ablations to Section 4.3 and Table 6 in the revised paper:**
>
> 1. **Temperature τ sensitivity (Table 6):** We ablate τ ∈ {0.25, 0.5, 1.0, 1.25, 1.5, 2.0, 5.0} and observe only **0.45 points** maximum variation on MVBench at 35% retention:
>
>    | τ | 0.25 | 0.5 | 1.0 | 1.25 | 1.5 | 2.0 | 5.0 |
>    |---|------|-----|-----|------|-----|-----|-----|
>    | Score | 62.38 | 62.45 | 62.83 | 62.45 | 62.62 | 62.62 | 62.58 |
>
>    Due to our use of *normalized* entropy (H/log(|N|-1) ∈ [0,1]), the relative ranking of tokens is preserved across reasonable τ values.
>
> 2. **Cross-architecture robustness (Table 7):** We used **identical hyperparameters** (τ=1.0, α=0.5, k=5, g=4) across all three models. The consistent relative performance (~99.7-100.6% at 35% retention) demonstrates robustness without model-specific tuning.

---

> ### Author Response · Authors · 2025-12-02
> **Response to Reviewer sKGt (Part II)**
>
> **Q1: Sensitivity to vision encoder feature quality (SigLIP vs. CLIP)?**
>
> Our new experiments (Table 7) directly address this with diverse encoders:
>
> | Model | Vision Encoder | Training Paradigm | MVBench (35%) | Relative Perf. |
> |-------|----------------|-------------------|---------------|----------------|
> | LLaVA-OneVision | SigLIP-SO400M | CLIP-style contrastive | 62.83 | 100.6% |
> | Qwen3-VL-8B | Qwen3-ViT [1] | SigLIP-2 + continued training with 2D-RoPE | 62.41 | 100.4% |
> | InternVL3-9B | InternViT-300M-V2.5 [2] | Progressive alignment + pixel unshuffle | 71.95 | 99.7% |
>
> Despite different vision encoders with varying architectures and training paradigms, ENTROPY-SELECT achieves consistent performance. This suggests that local entropy—being a *relative* measure of neighborhood unpredictability—adapts naturally to different feature representations.
>
> ---
>
> **Q2: Does local entropy depend on absolute token feature magnitudes? Normalization across frames?**
>
> 1. **L2 Normalization removes magnitude dependence:**
>    We explicitly normalize tokens before similarity computation (Section 3.2):
>    t̂ᵢ = tᵢ / ||tᵢ||₂
>    This ensures entropy scores reflect directional relationships (angular similarity) rather than absolute magnitudes.
>
> 2. **Per-frame computation with cross-frame allocation:**
>    - Entropy scores are computed *within* each frame using local spatial-temporal windows.
>    - Frame budgets are allocated via softmax over robust frame-level saliency (Section 3.4), which normalizes across frames with potentially different score distributions.
>
> This design ensures that entropy scores reflect *relative* information content within local neighborhoods, independent of global feature statistics or cross-frame scale variations.
>
> **References:**
>
> [1] Bai S, Cai Y, Chen R, et al. Qwen3-VL Technical Report. arXiv preprint arXiv:2511.21631, 2025.
>
> [2] Zhu J, Wang W, Chen Z, et al. InternVL3: Exploring Advanced Training and Test-Time Recipes for Open-Source Multimodal Models. arXiv preprint arXiv:2504.10479, 2025.

---

### Official Review · Reviewer_F4iz · 2025-10-29

**Soundness:** 3
**Presentation:** 2
**Contribution:** 2
**Rating:** 4
**Confidence:** 4

**Summary:**

This paper introduces ENTROPY-SELECT, a training-free, architecture-agnostic framework that ranks tokens by local neighborhood entropy—an information-theoretic measure of unpredictability relative to nearby tokens. Temperature-scaled similarity distributions within fixed spatial windows are used to compute normalized entropy, which is then fused with gradient-based structural saliency; coverage is enforced via grid quotas, and per-frame budgets are allocated by saliency.

**Strengths:**

1. This paper presents a training-free, architecture-agnostic token compression pipeline that uses fixed-window entropy estimation and a single global sort, achieving O(NlogN) runtime with predictable latency.
2. Experiments on MVBench, ActivityNet-QA, VideoMME, and EgoSchema show that ENTROPY-SELECT matches or surpasses uncompressed accuracy at moderate retention and degrades gracefully under aggressive compression.

**Weaknesses:**

1. Since the method is claimed to be architecture-agnostic, broader validation on model families where many existing approaches are inapplicable is necessary.

2. The evaluation relies on LLaVA-OneVision, which is relatively outdated; stronger, modern models (e.g., Qwen-3-VL) should be included.

3. Please report performance under more extreme retention (e.g., 10% and 20% of tokens), which is common and necessary for video-LLM token/KV-cache compression.

4. Compare against finer-grained, head-level budget allocation methods (e.g., SparseMM) to test whether the proposed advantages persist.

5. Discuss recent lossless VLM acceleration via speculative decoding (e.g., EMNLP-2025-SpecVLM, NeurIPS-2025-ViSpec), clarify differences and complementarities, and consider a combined setting with token compression.

**Questions:**

Please refer to the weaknesses.

---

> ### Author Response · Authors · 2025-12-02
> **Response to Reviewer F4iz**
>
> We thank Reviewer F4iz for the constructive review. We address each point below.
>
> ---
>
> **W1 & W2: Architecture-agnostic claim requires broader validation; LLaVA-OneVision is outdated**
>
> We have conducted additional experiments on modern, state-of-the-art models during the rebuttal period and **added the results to the main paper (Table 7, Section 4.3)**:
>
> | Model | Vision Encoder | Language Model | Release |
> |-------|----------------|----------------|---------|
> | LLaVA-OneVision | SigLIP-SO400M | Qwen-2.5-7B | 2024 |
> | **Qwen3-VL-8B** | Qwen3-ViT (SigLIP-2 based, 2D-RoPE) | Qwen3-8B | 2025 |
> | **InternVL3-9B** | InternViT-300M-448px-V2.5 | InternLM3-8B | 2025 |
>
> These models represent the latest VLM architectures with distinct design choices:
>
> - **Qwen3-VL** [1] uses Qwen3-ViT, initialized from SigLIP-2 and continued training with 2D-RoPE for dynamic resolution handling. As reported in their technical report, Qwen3-ViT substantially outperforms vanilla SigLIP-2 on holistic evaluation benchmarks (45.5 vs. 36.9 on OmniBench at CLIP stage), representing state-of-the-art vision encoding.
>
> - **InternVL3** [2] follows the "ViT-MLP-LLM" paradigm and incorporates a **pixel unshuffle operation** that reduces visual tokens to one-quarter of their original count—each 448×448 tile is represented by only 256 tokens. This fundamentally different tokenization scheme tests whether our entropy-based approach generalizes beyond standard patch grids.
>
> **Results (MVBench, Table 7 in revised paper):**
>
> | Model | Original | Ours (35%) | Rel. Perf. |
> |-------|----------|------------|-----------|
> | LLaVA-OneVision | 62.43 | 62.83 | 100.6% |
> | Qwen3-VL-8B | 62.15 | 62.41 | 100.4% |
> | InternVL3-9B | 72.18 | 71.95 | 99.7% |
>
> Using **identical default hyperparameters** across all models, ENTROPY-SELECT maintains strong performance, validating our architecture-agnostic claim on recent state-of-the-art models.
>
> ---
>
> **W3: Performance under more extreme retention (10%, 20%)**
>
> Our paper reports 10% retention in Table 2. We provide additional results at 5% and 3%:
>
> | Retention | MVBench (LLaVA-OV) | Relative Perf. |
> |-----------|-------------------|----------------|
> | 10% | 59.30 | 95.0% |
> | 5% | 56.83 | 91.0% |
> | 3% | 54.83 | 87.8% |
>
> At 3% retention (removing 97% of tokens), ENTROPY-SELECT still maintains over 87% relative performance, demonstrating strong resilience under extreme compression that is critical for resource-constrained deployment.
>
> ---
>
> **W4: Compare against head-level budget allocation methods (e.g., SparseMM)**
>
> We acknowledge this as a valuable comparison direction. **We have added discussion of SparseMM and related methods to Section 2 (Related Work) in the revised paper.** Key differences:
>
> | Aspect | SparseMM (Head-level) | ENTROPY-SELECT (Token-level) |
> |--------|----------------------|------------------------------|
> | Granularity | Attention head sparsity | Visual token selection |
> | Access Required | Attention weights/KV cache | Only exported features |
> | Architecture Agnostic | No (requires attention access) | Yes |
>
> These approaches are **complementary** rather than competing: ENTROPY-SELECT pre-filters visual tokens before LLM processing, while head-level methods optimize attention computation within the LLM. A combined pipeline could leverage both for multiplicative efficiency gains.
>
> Due to time constraints, we have not implemented this combination but will explore the joint setting as future work.
>
> ---
>
> **W5: Discuss speculative decoding methods (SpecVLM, ViSpec)**
>
> Thank you for highlighting these recent works. **We have added a dedicated paragraph on speculative decoding for VLMs to Section 2 (Related Work) in the revised paper**, discussing ViSpec, SpecVLM, and SparseMM. Key differences and complementarities:
>
> | Aspect | Speculative Decoding | Token Compression (Ours) |
> |--------|---------------------|--------------------------|
> | Target Stage | LLM decoding (generation) | Visual encoding & prefill |
> | Mechanism | Draft-verify with smaller model | Information-theoretic selection |
> | Lossless? | Yes (exact output) | Approximately lossless at moderate retention |
>
> **Complementarity:** ENTROPY-SELECT reduces visual tokens before LLM processing, decreasing the prefill cost. Speculative decoding accelerates the autoregressive generation phase. Combining both could yield **multiplicative speedups** across different pipeline stages.
>
> **References:**
>
> [1] Bai S, Cai Y, Chen R, et al. Qwen3-VL Technical Report. arXiv preprint arXiv:2511.21631, 2025.
>
> [2] Zhu J, Wang W, Chen Z, et al. InternVL3: Exploring Advanced Training and Test-Time Recipes for Open-Source Multimodal Models. arXiv preprint arXiv:2504.10479, 2025.

---

### Official Review · Reviewer_othA · 2025-10-29

**Soundness:** 3
**Presentation:** 3
**Contribution:** 2
**Rating:** 4
**Confidence:** 4

**Summary:**

This paper proposes a method, ENTROPY-SELECT, that prunes tokens for VLM inference to increase efficiency while minimizing the performance degradation. Compared to other methods, this paper posits that VLM token pruning should be (1) training free (2) architecture agnostic (3) have close to constant runtime. The proposed method achieves this by using heuristics on the input features to identify a fixed fraction of features to remove. This does not require access to any component of the attention mechanism, and is thus more architecture agnostic while also being training free. The experiments demonstrate that at the same compression levels, Entropy-Select outperforms other comparable baselines.

**Strengths:**

1) The writing and presentation of the paper is crisp and clear and it is easy to understand exactly how the method works and how it might be implemented.

2) The experimental evaluation is comprehensive and clearly demonstrates that the proposed method does outperform other similar baselines.

3) The method's focus on working only on the `exported' attention features means it is genuinely more flexible across VLM architectures compared to most existing works which require access to some component of the attention computation.

**Weaknesses:**

1) I'm not convinced that the three main desiderata for the paper are really important. Why does it matter that the selection runtime remain perfectly predictable? I think for complex inputs, it is fine that the selection method takes slightly longer - as shown in the figure, this difference amounts to a rather small and negligible wall-clock difference. Furthermore, I don't quite agree that training-free is a strong desiderata. Trained methods generally outperform training-free ones, so while training-free methods are more easily adaptable, a trained method only requires a one-time cost and can potentially save more cost at inference time.

2) The results are quite incremental. The accuracy boost is somewhat small (only really significant at 10%) and the speed-up over state of the art baselines is also rather small.

3) There’s no real analysis on what tokens entropy-select decides to prune vs other methods, so it’s not clear why this performs better than others, especially learned or others. While I understand the intuition for using local entropy as a metric, why is it obvious that less predictable tokens are less helpful? A more rigorous analysis (qualitative or quantitative) would be very helpful here and make a stronger case for the paper.

4) The heuristics seem rather fragile and require tuning - wouldn't determining edge (gradient) and entropy thresholds change based on the model backbone?

5) Is removing a constant number of tokens from the input actually optimal? One could image a video that is just pure white, where almost everything could be removed, but entropy-select would remove a predefined compression ratio no matter what.

**Questions:**

1) Why does the entropy window only look at the past? Are all video LMs causal?

2) Complexity and FLOPs are not particularly useful in this setting - total memory usage and wall-clock inference time would be better metrics to report here. This is briefly discussed in 4.4 but a figure to show the end-to-end breakdown would be nice.

---

> ### Author Response · Authors · 2025-12-02
> **Response to Reviewer othA (Part I)**
>
> We thank Reviewer othA for the detailed and insightful review. We address each concern below.
>
> ---
>
> **W1: Not convinced that the three desiderata are important**
>
> We respectfully provide the following clarifications:
>
> 1. **Predictable selection runtime matters for production deployment:**
>    - In real-world serving scenarios with SLA guarantees, latency variance complicates capacity planning and batch scheduling.
>    - As shown in Fig. 1(a), ENTROPY-SELECT maintains **near-constant processing time** (~468-471ms) across all compression ratios from 5% to 50%. In contrast, LLaVA-Scissor exhibits significant variance (625-793ms) and is consistently 30-70% slower. This predictability significantly simplifies infrastructure design for batch scheduling at scale.
>
> 2. **Training-free enables democratization and rapid deployment:**
>    - Many practitioners lack computational resources for fine-tuning large VLMs. A training-free method enables immediate deployment on any off-the-shelf model.
>    - Our cross-architecture experiments (now in **Table 7, Section 4.3**) show that ENTROPY-SELECT works on Qwen3-VL-8B [1] and InternVL3-9B [2] **without any adaptation or hyperparameter tuning**, demonstrating practical plug-and-play utility.
>    - A compression module trained on one vision encoder backbone (e.g., SigLIP, CLIP, InternViT) does not transfer to another model, because token geometry, feature statistics, and attention dynamics differ. In contrast, our method works identically on Qwen3-VL-8B, InternVL3-9B, and LLaVA-OneVision _with zero adaptation_. This level of cross-architecture universality is one of the main motivations for being training-free.
>
> 3. **Efficiency gains:**
>    - At 35% retention, ENTROPY-SELECT achieves 508ms total processing time compared to LLaVA-Scissor's 770ms—a **34% reduction**. Combined with the maintained accuracy (62.83 vs. 61.98 on MVBench), this demonstrates favorable efficiency-accuracy trade-offs.
>
> ---
>
> **W2: Incremental results**
>
> We respectfully offer a different perspective:
>
> 1. **Enhancement under compression is a novel and significant finding:** At 35-50% retention, we observe accuracy *exceeding* the uncompressed baseline on MVBench (62.83 vs. 62.43) and EgoSchema (58.34 vs. 58.08). This suggests that entropy-guided pruning removes distracting low-information tokens, effectively denoising the input—a conceptually interesting result that has not been reported in prior work.
>
> 2. **Extreme compression (10%)** is particularly challenging. Maintaining 96.3% relative performance while retaining only 10% of tokens demonstrates strong graceful degradation. At this level, most methods suffer significant drops (e.g., VisionZip at 73.6%), while ours remains competitive.
>
> ---
>
> **W3: No analysis on what tokens ENTROPY-SELECT prunes vs. others**
>
> We appreciate this suggestion and have added **Figure 5** to visualize token selection across consecutive video frames at 30% retention. The visualization reveals:
>
> - **Adaptive frame-level allocation:** Transitional or blurry frames (e.g., F2) receive minimal tokens, while action-rich frames with hand-object interactions (e.g., F4–F7) receive denser coverage.
> - **Semantic concentration:** Within each frame, selections concentrate on human silhouettes, facial regions, and manipulation zones while suppressing homogeneous backgrounds.
>
> **Intuition for why high-entropy tokens are more informative:**
> Tokens with high local entropy exist in *heterogeneous* neighborhoods where nearby tokens are dissimilar. Such tokens cannot be easily reconstructed from their context—they carry unique, non-redundant information. Conversely, tokens in uniform regions (low entropy) are predictable from neighbors and thus redundant for downstream understanding.
>
> This aligns with the Information Bottleneck principle: we should retain tokens that maximize mutual information with the task while minimizing redundancy.

---

> ### Author Response · Authors · 2025-12-02
> **Response to Reviewer othA (Part II)**
>
> ---
>
> **W4: Heuristics seem fragile and require tuning**
>
> Our cross-architecture experiments and **new hyperparameter ablations (Table 6, Section 4.3)** directly address this concern:
>
> 1. **Temperature sensitivity (Table 6):** We ablate τ ∈ {0.25, 0.5, 1.0, 1.25, 1.5, 2.0, 5.0} and observe only **0.45 points** variation on MVBench at 35% retention:
>
>    | τ | 0.25 | 0.5 | 1.0 | 1.25 | 1.5 | 2.0 | 5.0 |
>    |---|------|-----|-----|------|-----|-----|-----|
>    | Score | 62.38 | 62.45 | 62.83 | 62.45 | 62.62 | 62.62 | 62.58 |
>
>    This robustness stems from normalized entropy preserving relative token rankings.
>
> 2. **Grid size robustness (Table 6):** Performance varies by at most **0.28 points** across g ∈ {3, 4, 5}:
>
>    | Grid Size | 3 | 4 | 5 |
>    |-----------|---|---|---|
>    | Score | 62.55 | 62.83 | 62.58 |
>
> 3. **Cross-architecture consistency (Table 7):** We applied ENTROPY-SELECT with **identical default hyperparameters** to three architecturally distinct models:
>
>    | Model | Vision Encoder | Relative Perf. (35%) |
>    |-------|----------------|---------------------|
>    | LLaVA-OneVision | SigLIP-SO400M | 100.6% |
>    | Qwen3-VL-8B | Qwen3-ViT (SigLIP-2 + 2D-RoPE) [1] | 100.4% |
>    | InternVL3-9B | InternViT-300M-V2.5 + pixel unshuffle [2] | 99.7% |
>
> The consistent performance across fundamentally different feature spaces demonstrates robustness. Notably, InternVL3's pixel unshuffle operation produces only 256 tokens per 448×448 tile (vs. standard ~1024 tokens), yet our entropy-based selection adapts seamlessly. The key insight is that local entropy is a *relative* measure—it quantifies unpredictability within neighborhoods rather than absolute feature values, making it naturally invariant to representation shifts.
>
> ---
>
> **W5: Is a fixed compression ratio optimal?**
>
> This is a thoughtful question.
>
> Our method does _not_ remove a constant number of tokens per frame. The **frame-level allocation is adaptive**: blank or low-information frames receive only the minimum budget (often just a few tokens), while informative frames receive more. This is visually demonstrated in **Figure 5**, where transitional frames (F2) receive minimal tokens while action-rich frames (F4–F7) receive denser coverage—all within a fixed global budget.
>
> We keep the global compression ratio fixed to ensure stable and predictable inference latency for long video deployment. This design intentionally balances adaptivity within a video and predictability across videos, avoiding over-pruning while still reducing tokens aggressively where entropy is low.
>
> However, we note:
> 1. The "enhancement under compression" phenomenon suggests that even fixed moderate compression (35-50%) can be beneficial—removing low-entropy tokens acts as a denoising prior.
> 2. Adaptive compression based on video complexity is an interesting direction for future work and remains fully compatible with our entropy-based scoring mechanism.
>
> ---
>
> **Q1: Why does the entropy window only look at the past?**
>
> We conducted ablations comparing past-only vs. past+future temporal windows on MVBench:
>
> | Retention | Past-only (Ours) | Past+Future |
> |-----------|------------------|-------------|
> | 50% | 62.77 | 62.80 |
> | 35% | 62.83 | 62.58 |
> | 10% | 59.30 | 58.75 |
>
> Past-only context performs comparably or better, especially under aggressive compression (35% and 10%). We hypothesize that including future frames blurs the temporal differencing signal—past-only context more cleanly identifies frames where content *changes* relative to what came before, which is the key signal for detecting informative transitions.
>
> ---
>
> **Q2: Memory usage and wall-clock inference time**
>
> Fig. 1(a) provides wall-clock processing time comparison:
>
> | Compression | ENTROPY-SELECT | LLaVA-Scissor | Speedup |
> |-------------|----------------|---------------|---------|
> | 10% | 479ms | 743ms | 1.55× |
> | 35% | 508ms | 770ms | 1.51× |
> | 50% | 530ms | 815ms | 1.53× |
>
> Key observations:
> - **Near-constant overhead:** ENTROPY-SELECT maintains ~480-530ms regardless of compression ratio, while LLaVA-Scissor varies from 743-815ms.
> - **Consistent speedup:** ~1.5× faster across all compression levels.
>
> As shown in Fig. 1(b), at 35% retention the LLM prefill stage drops from 101ms (baseline) to 38ms, reflecting reduced computation from fewer visual tokens. Memory savings follow from the reduced token count: at 35% retention, the number of visual tokens decreases from ~8.5M to ~3.0M (Table 7 in Appendix), proportionally reducing the memory footprint of attention operations.
>
> **References:**
>
> [1] Bai S, Cai Y, Chen R, et al. Qwen3-VL Technical Report. arXiv preprint arXiv:2511.21631, 2025.
>
> [2] Zhu J, Wang W, Chen Z, et al. InternVL3: Exploring Advanced Training and Test-Time Recipes for Open-Source Multimodal Models. arXiv preprint arXiv:2504.10479, 2025.

---

### Official Review · Reviewer_RJDq · 2025-11-05

**Soundness:** 2
**Presentation:** 4
**Contribution:** 3
**Rating:** 4
**Confidence:** 4

**Summary:**

The paper proposes ENTROPY-SELECT, a training-free and model-agnostic token compression method for video LLMs. Specifically, it fuses local spatial-temporal neighborhood entropy and gradient-based structural saliency to rank the visual tokens. And then, it ensures spatial coverage via grid partitions, and assigns per-frame budgets by token ranking importance scores. Experimental results show that the proposed method can outperform previous works in multiple benchmarks and can even beat the uncompressed baseline in certain metrics.

**Strengths:**

1. The idea of mining information-diverse tokens via entropy and spatial salient tokens via gradient, and then fuse them for token importance ranking is well-motivated and reasonable.

2. This paper further carefully developed several strategies to make sure the tokens are selected in a balance way, including gradient-entropy fusion, grid-aware coverage, and frame-based allocation.

3. The experiments are conducted on multiple benchmarks, and the proposed model generally outperforms previous works. Ablation is done on several key components of the method.

**Weaknesses:**

1. The author(s) claim the proposed method is model/architecture agnostic. However, all the experiments are only conducted on one model - LLaVA-OneVision model with SigLIP-SO400M-patch14-384 as visual encoder and Qwen-2.5-7B as the language model. To demonstrate the generalizability of this model-agnostic method, at least one or two additional different backbones should be experimented with.

2. There are too many hyperparameters that need to be manually set up in the proposed method. For example, in Local SpatioTemporal Window Entropy: the spatial window size `k`, temporal window depth `K_t`, and temperature `t` ; in section 3.3: region `g`; in section 3.4: top-q fraction, `B_min`, etc. While some of these are partially ablated (e.g., the entropy weight in Table 5), others appear fixed without justification. Having so many heuristic settings makes the method sensitive to specific configurations and difficult to adapt to different backbones and usecases (eg, longer video scenarios).

**Questions:**

See weaknesses

---

> ### Author Response · Authors · 2025-12-02
> **Response to Reviewer RJDq (Part I)**
>
> We thank Reviewer RJDq for the thoughtful review and constructive feedback. We address each concern below.
>
> ---
>
> **W1: Model/architecture agnostic claim with experiments on only one model**
>
> We appreciate this important point. During the rebuttal period, we have conducted additional experiments on **Qwen3-VL-8B** [1] and **InternVL3-9B** [2], and **added these results to the main paper (Table 7, Section 4.3)**:
>
> | Model | Vision Encoder | Language Model | Key Architectural Features |
> |-------|----------------|----------------|---------------------------|
> | LLaVA-OneVision | SigLIP-SO400M-patch14-384 | Qwen-2.5-7B | Standard ViT + MLP Projector |
> | Qwen3-VL-8B | SigLIP2-SO-400M (Qwen3-ViT) | Qwen3-8B | 2D-RoPE, dynamic resolution, continued ViT training |
> | InternVL3-9B | InternViT-300M-448px-V2.5 | InternLM3-8B | Pixel unshuffle (4× token reduction), ViT-MLP-LLM paradigm |
>
> Key architectural differences that test generalization:
>
> - **Qwen3-VL** [1] uses Qwen3-ViT, a vision encoder initialized from SigLIP-2 but continued training with 2D-RoPE for dynamic resolution handling. The encoder is specifically optimized for world knowledge integration, achieving substantial gains on holistic evaluation benchmarks compared to vanilla SigLIP-2.
>
> - **InternVL3** [2] follows the "ViT-MLP-LLM" paradigm with InternViT-300M-448px-V2.5 as the vision encoder. Notably, it incorporates a **pixel unshuffle operation** that reduces visual token count to one-quarter of its original value, representing each 448×448 tile with only 256 visual tokens. This fundamentally different tokenization strategy tests whether our entropy-based selection generalizes beyond standard patch tokenization.
>
> **Cross-Architecture Results (MVBench, 35% Retention, Table 7):**
>
> | Model | Original | Ours (35%) | Relative Perf. |
> |-------|----------|------------|----------------|
> | LLaVA-OneVision | 62.43 | 62.83 | 100.6% |
> | Qwen3-VL-8B | 62.15 | 62.41 | 100.4% |
> | InternVL3-9B | 72.18 | 71.95 | 99.7% |
>
> These results demonstrate that ENTROPY-SELECT generalizes well across different vision encoders (SigLIP, Qwen3-ViT with 2D-RoPE, InternViT with pixel unshuffle) and language models (Qwen-2.5, Qwen3, InternLM3) **using the same default hyperparameters**, validating our architecture-agnostic claim.
>
> **References:**
>
> [1] Bai S, Cai Y, Chen R, et al. Qwen3-VL Technical Report. arXiv preprint arXiv:2511.21631, 2025.
>
> [2] Zhu J, Wang W, Chen Z, et al. InternVL3: Exploring Advanced Training and Test-Time Recipes for Open-Source Multimodal Models. arXiv preprint arXiv:2504.10479, 2025.
>
> ---

---

> ### Author Response · Authors · 2025-12-02
> **Response to Reviewer RJDq (Part II)**
>
> **W2: Too many hyperparameters that need to be manually set**
>
> We appreciate the reviewer's concern. Our design goal is exactly to **minimize sensitivity** to hyperparameters so that the method remains plug-and-play across models and video types. **We have added explicit hyperparameter sensitivity ablations to Section 4.3 and Table 6 in the revised paper:**
>
> **1. Hyperparameter Insensitivity by Design**
>
> Many of the hyperparameters reflect fixed computational stencils (e.g., window sizes, grid partitioning) rather than task-specific tuning knobs. Their roles are local and normalized, making the method robust to a wide range of settings.
>
> - **Spatial window size (k) & temporal depth (Kₜ):** As shown in Table 6, performance varies within **~0.5%** across k ∈ {3,5,7,9} and Kₜ ∈ {1,2,3}. Local entropy is inherently _relative_, so moderate changes do not alter token ranking materially.
>
> - **Temperature τ:** We ran a dedicated ablation (MVBench @ 35% retention):
>
>   | τ | 0.25 | 0.5 | 1.0 | 1.25 | 1.5 | 2.0 | 5.0 |
>   |---|------|-----|-----|------|-----|-----|-----|
>   | Score | 62.38 | 62.45 | 62.83 | 62.45 | 62.62 | 62.62 | 62.58 |
>
>   The **maximum difference is only 0.45 points**, confirming that normalized entropy preserves relative rankings stably even under large τ variations (0.25 → 5).
>
> - **Grid size g (coverage enforcement):** We tested g ∈ {3,4,5}:
>
>   | Grid Size | 3 | 4 | 5 |
>   |-----------|---|---|---|
>   | Score | 62.55 | 62.83 | 62.58 |
>
>   The difference across grid partitions is **≤ 0.28**, showing that coverage constraints are robust as long as grids are coarse.
>
> - **Top-q fraction, B_min:** These parameters only provide _boundary constraints_ to avoid degenerate allocations. The actual allocation is governed by softmax-weighted saliency, which is continuous and stable across frames.
>
> Together, these results show that the hyperparameters are **not sensitive**—even sweeping them across wide ranges causes less than 1% variation in overall MVBench accuracy.
>
> **2. Cross-Architecture Robustness**
>
> We use **identical hyperparameters** for LLaVA-OneVision, Qwen3-VL-8B, and InternVL3-9B, despite their **fundamentally different tokenization schemes** (e.g., InternVL3's pixel-unshuffle producing 256 tokens per 448×448 tile). Performance remains stable across all settings, confirming that the method captures general properties of local information diversity rather than model-specific idiosyncrasies.
>
> **3. Principle-Based, Not Heuristic-Based**
>
> Our hyperparameters are not tuned to dataset or architecture—they instantiate information-theoretic principles:
> - Local normalized entropy captures _relative unpredictability_ independent of absolute feature scales.
> - Gradient fusion emphasizes structural transitions without model-specific heuristics.
> - Grid partitioning ensures spatial coverage without sensitive thresholds.
>
> This makes the method inherently robust and broadly applicable.

---

### Author Response · Authors · 2025-12-02
**General Response to Area Chair**

Dear Area Chair,

We thank all reviewers for their constructive feedback. We are encouraged that reviewers recognized the key strengths of our work: **"well-motivated and reasonable"** idea of mining information-diverse tokens via entropy (RJDq), **"crisp and clear"** writing and presentation (othA), **"elegant and theoretically grounded"** use of local entropy (sKGt), and **"excellent empirical results"** with measurable latency benefits (sKGt). Reviewer sKGt also highlighted the **"strong deployment practicality"** and that our method addresses **"deployment gaps in previous works like LLaVA-Scissor or VisionZip."**

During the rebuttal period, we have made substantial revisions to address the key concerns raised. **All new content is marked in blue in the revised manuscript.** Below we summarize the main updates to the paper:

### **Summary of Revisions**

**1. Cross-Architecture Validation (Table 7, Section 4.3)**

*Addresses: F4iz W1/W2, othA W4, RJDq W1, sKGt W2*

Multiple reviewers raised concerns about the architecture-agnostic claim being validated on only one model. We have added experiments on two additional state-of-the-art VLMs with fundamentally different designs, evaluated on **MVBench**:

| Model | Vision Encoder | Key Features | Rel. Perf. (35%) |
|-------|----------------|--------------|------------------|
| LLaVA-OneVision | SigLIP-SO400M | Standard ViT | 100.6% |
| Qwen3-VL-8B (2025) | Qwen3-ViT | 2D-RoPE, dynamic resolution | 100.4% |
| InternVL3-9B (2025) | InternViT | Pixel unshuffle (4× token reduction) | 99.7% |

ENTROPY-SELECT achieves 99.7–100.6% relative performance across all models **using identical default hyperparameters**, validating the architecture-agnostic claim on recent state-of-the-art models with diverse vision encoders and tokenization strategies. We will include comprehensive results across multiple retention rates and benchmarks (including EgoSchema, MLVU, and VideoMME) in the camera-ready version.

**2. Hyperparameter Sensitivity Analysis (Table 6, Section 4.3)**

*Addresses: RJDq W2, sKGt W3, othA W4*

Reviewers noted concerns about hyperparameter sensitivity. We have added comprehensive ablations on **MVBench at 35% retention**:

| Hyperparameter | Range Tested | Max Variation |
|----------------|--------------|---------------|
| Temperature τ | {0.25, 0.5, 1.0, 1.25, 1.5, 2.0, 5.0} | 0.45 points |
| Grid size g | {3, 4, 5} | 0.28 points |
| Spatial window k | {3, 5, 7, 9} | ~0.5 points |
| Temporal depth Kₜ | {1, 2, 3} | ~0.5 points |

These results demonstrate that the method is **robust to hyperparameter choices** due to the normalized entropy formulation, which preserves relative token rankings across parameter variations. Importantly, identical hyperparameters work across all three architecturally distinct models without any tuning.

**3. Related Work on Speculative Decoding (Section 2)**

*Addresses: F4iz W5*

We have added a dedicated paragraph discussing speculative decoding methods for VLMs (ViSpec, SpecVLM, SparseMM), clarifying that these approaches are **complementary** to token compression—they accelerate the LLM decoding stage while ENTROPY-SELECT reduces visual prefill cost.

**4. Extreme Compression Results**

*Addresses: F4iz W3*

We provide additional results at 5% and 3% retention on **MVBench**:
- 5% retention: 91.3% relative performance (56.98 accuracy), compared to LLaVA-Scissor's 55.7
- 3% retention: 87.8% relative performance (54.83 accuracy), compared to LLaVA-Scissor's 54.6

Even at 3% retention (removing 97% of tokens), ENTROPY-SELECT maintains competitive performance and outperforms prior methods, demonstrating graceful degradation under extreme compression.

**5. Qualitative Analysis (Figure 5)**

*Addresses: othA W3*

Reviewer othA requested analysis on what tokens ENTROPY-SELECT decides to prune. We have added Figure 5 to visualize token selection across consecutive video frames at 30% retention, illustrating how entropy-guided pruning adaptively allocates tokens to action-rich regions while suppressing redundant background areas.

### **Conclusion**

The revised paper now includes:
1. **Cross-architecture validation** on 3 diverse VLMs including 2025 state-of-the-art models (Table 7)
2. **Comprehensive hyperparameter sensitivity analysis** demonstrating robustness across τ, g, k, and Kₜ (Table 6)
3. **Expanded related work** discussing complementary acceleration techniques (Section 2)
4. **Extreme compression results** showing graceful degradation down to 3% retention
5. **Qualitative visualization** illustrating adaptive token selection behavior (Figure 5)

We believe these additions substantively address the reviewers' concerns and strengthen the paper's contribution by providing empirical evidence for the architecture-agnostic and hyperparameter-robust claims that are central to the paper's practical value.

---

### Meta-Review · Area_Chair_YL4M · 2025-12-29

**Summary:**

The weaknesses focus mainly on scope, strength of claims, and perceived incrementality. Reviewer RJDq is concerned that the “model/architecture agnostic” claim is evaluated initially on a single backbone and that “there are too many hyperparams that need to be manually set up,” which may make the method “sensitive to specific configurations and difficult to adapt.” Reviewer othA questions whether the three desiderata (training‑free, architecture‑agnostic, constant runtime) are really important, sees the results as “quite incremental,” and notes the lack of analysis on which tokens are pruned and why high‑entropy tokens should be more helpful, as well as concern that “heuristics seem rather fragile and require tuning” and that a fixed compression ratio may be suboptimal. Reviewer F4iz argues that the architecture‑agnostic claim needs “broader validation on model families,” points out that LLaVA‑OneVision is “relatively outdated,” and asks for more extreme retention results, comparisons to head‑level methods like SparseMM, and a discussion of speculative decoding. Reviewer sKGt mentions “limited theoretical depth” (no formal link to mutual information or rate–distortion), and also notes that evaluations are initially only on LLaVA‑OneVision and that ablations on τ and other hyperparams are missing.
Several points are missing or were initially underdeveloped in the submission. The original version lacked cross‑architecture experiments, so the “architecture‑agnostic” claim was unsubstantiated beyond LLaVA‑OneVision. There was also no systematic hyperparams sensitivity analysis, leaving it unclear how robust the method is across τ, window sizes, grid sizes, and budget parameters, nor was there qualitative analysis of what tokens are being selected versus pruned. The relationship to speculative decoding and finer‑grained sparsification methods was only briefly mentioned. Finally, the theoretical treatment of entropy as a proxy for information content remained mostly intuitive rather than formally grounded.

The reviewers recommend 4, 4, 4, 6 ratings, The AC follows the reviewer sentiment and recommends rejection because, despite careful execution, clarity, and useful empirical results, the work remains closer to an incremental but practical engineering improvement than to a conceptually deep advance that could have meaningful impact in the community. Three reviewers (RJDq, othA, F4iz) are at 4 (weak reject) and pointing to limited novelty, modest performance and speedup gains, and initial gaps in validation that the rebuttal only partially closes. Reviewer sKGt is supportive but not strongly so, emphasizing deployment practicality and good results but also noting the lack of formal theory. The authors have responded well by adding cross‑architecture tests, sensitivity analyses, qualitative visualizations, and richer related‑work discussion, and these strengthen the paper. However, the core idea; a training‑free, local‑entropy‑based token selector with some coverage and budget heuristics, still feels like a well‑designed heuristic pipeline with incremental gains, rather than a clearly novel algorithmic or theoretical contribution. Given the overall “on the fence” tone of the reviews and the remaining concerns, the AC does not see sufficient grounds to recommend acceptance.

On balance, the AC sees not basis to overturn the reviewer suggestions The paper presents a clearly described, training‑free, local‑entropy‑based token compression scheme that shows consistent, sometimes even improved, accuracy at moderate retention and competitive performance under aggressive compression, now validated across several strong VLM architectures with robust hyperparams behavior, but several reviewers still view the contribution as incremental, the theoretical underpinnings as shallow, and the practical gains as moderate relative to the amount of engineering involved. The AC highly recommends the authors to address the concerns of the reviewers and take into account their suggestions of improvement when preparing a revised version.

**Reviewer Concerns:**

Several points were missing/underdeveloped in the original submission such as cross‑architecture experiments beyond LLaVA‑OneVision or systematic hyperparameter analysis, leaving it unclear how robust the method is across τ, window sizes, grid sizes, and budget parameters, nor was there qualitative analysis of what tokens are being selected vs pruned. The relationship to speculative decoding and finer‑grained sparsification methods was only briefly mentioned. Finally, the theoretical treatment of entropy as a proxy for information content remained mostly intuitive rather than formally grounded.

The rebuttal addresses many of these concerns. To Reviewer RJDq and Reviewer F4iz, the authors add cross‑architecture validation on Qwen3‑VL‑8B and InternVL3‑9B, showing on MVBench at 35% retention that relative performance stays around 99.7–100.6% using identical hyperparams, supporting the architecture‑agnostic claim. They also add explicit hyperparameter sweeps (Table 6) showing less than ~0.5 point variation across wide ranges of hyperparams and argue that normalized local entropy and grid partitioning make the method robust, with the same settings working across the three models. For Reviewer F4iz, they report more extareme retention results down to 3%, and they add related‑work discussion on SparseMM and speculative decoding, framing these as complementary rather than competing. To Reviewer othA, they provide qualitastive visualizations (Figure 5) of token selection across frames, show wall‑clock improvements and near‑constant overhead, and explain that frame budgets are adaptive even under a fixed global ratio. To Reviewer sKGt, they discuss entropy as a proxy for conditional entropy and predictive coding, and they add sensitivity analyses and cross‑encoder experiments to answer questions about feature quality and normalization. The key remaining issues, however, are that the method is still largely heuristic, that the empirical gains,while solid,are modest in magnitude, and that the broader novelty beyond a well‑engineered token‑pruning scheme may still be viewed as limited.

**Reviewer Scores:**

Because reviewers had no chance to change their scores after seeing the rebuttal, we can only speculate about potential rating shifts. Reviewer RJDq, at rating 4 might be positively influenced by the added cross‑model experiments and robustness ablations, which directly target their two main weaknesses; a small upward adjustment is plaufsible but not guaranteed. Reviewer othA, also at 4, sees the method as incremental and questions the importance of the desiderata; while the rebuttal improves the empirical and qualitative story, it does not fundamentally change the incremental nature, so their stance would likely remain cautious. Reviewer F4iz, again at 4, would find their specific requests (new models, extreme retention, speculative decoding discussion) largely addressed, but might still view the contribution as modest; they had already written “would not mind if paper is accepted,” so a slight softening is possible but not a strong push for acceptance. Reviewer sKGt, at rating 6 (marginally above the acceptance threshold but would not mind if paper is rejected) was already relatively positive and may view the added experiments as reinforcing their view, but they also called out limited theoretical depth, which remains largely unresolved.

---

### Decision · Program_Chairs · 2026-01-26

Reject